# The importance of acid processed meteoric smoke relative to meteoric fragments for crystal nucleation in polar stratospheric clouds

Alexander D. James[1], Finn Pace[1], Sebastien N. F. Sikora[2], Graham W. Mann[2], John M. C. Plane[1], Benjamin J. Murray[2]

[1]School of Chemistry, University of Leeds, Leeds, LS9 2JT, UK
[2]School of Earth and Environment, University of Leeds, Leeds, LS9 2JT, UK

*Correspondence to*: Alexander D. James (A.James1@leeds.ac.uk) and Benjamin J. Murray (B.J.Murray@leeds.ac.uk)

**Abstract.** Nitric Acid Trihydrate (NAT) crystal formation in the absence of water ice is important for a subset of Polar Stratospheric Clouds (PSCs) and thereby ozone depletion. It has been suggested that either fragmented meteoroids or meteoric smoke particles (MSPs), or possibly both, are important as heterogeneous nuclei of these crystals. Previous work has focused on the nucleating ability of meteoric material in nitric acid in the absence of sulfuric acid. However, it is known that when immersed in stratospheric sulfuric acid droplets, metal-containing meteoric material particles partially dissolve and components can re-precipitate as silica and alumina that have different morphologies to the original meteoric material. Hence, in this study we experimentally and theoretically explore the relative role that sulfuric acid-processed MSPs and meteoric fragments may play in NAT nucleation in PSCs.

We compared meteoric fragments that had been recently prepared (by milling a meteorite sample) to a sample annealed under conditions designed to simulate heating during entry into the Earth's atmosphere. Whilst the addition of sulfuric acid decreased the nucleating ability of the recently milled meteoric material relative to nucleation in binary nitric acid-water solutions (at similar NAT saturation ratio), the annealed meteoric fragments nucleated NAT with a similar effectiveness in both solutions. However, combining our results with measured fluxes of meteoric material to the Earth, sedimentation modelling and recent experiments on fragmentation of incoming meteoroids, suggests that there are unlikely to be sufficient fragments to contribute to the nucleation of crystalline NAT particles.

We then considered silica formed from sulfuric acid processed MSPs. Our previous work showed that nano-particulate silica (radius ~6 nm) is a relatively poor promoter of nucleation compared with micron scaled silica particles, which were more effective. Both materials have similar chemical and structural (crystallographically amorphous) properties, indicating size is critical. Here we account for surface curvature of primary grains using Classical Nucleation Theory (CNT) to explore this size dependence. This model is able to explain the discrepancy in nucleation effectiveness of fumed silica and fused quartz, by treating their nucleating activity (contact angle) as equal but with differing particle size (or surface curvature), assuming interfacial energies that are physically reasonable. Here we use this CNT model to present evidence that nucleation of NAT

on acid processed MSPs, where the primary grain size is 10s nm, is also effective enough to contribute to NAT crystals in early season PSCs where there is an absence of ice.

This study demonstrates that modelling of crystal nucleation in PSCs and resulting ozone depletion relies on accurate understanding of the transport and chemical processing of MSPs. This will affect estimated sensitivity of stratospheric chemistry to rare events such as large volcanic eruptions and long-term forecasting of ozone recovery in a changing climate.

**1 Introduction**

With record ozone loss observed in the Arctic winter 2019-2020 (Lawrence et al., 2020;Dameris et al., 2021;Manney et al., 2020;Wohltmann et al., 2020), it is increasingly clear that understanding the chemistry which occurs in the winter polar vortex is important for predicting future recovery of polar ozone. Aerosol science, and nucleation of crystalline components of Polar Stratospheric Clouds (PSCs) in particular, remains a key uncertainty in modelling chlorine and bromine activation and ozone

destruction. Nucleation is particularly important because the crystallisation of nitric acid hydrates (NAX) and water ice affects both the total amount and the kinetics of ozone destroying species activation of the heterogeneous surface of PSC particles (Brakebusch et al., 2013;Wegner et al., 2012). The growth and sedimentation of these nitric acid containing particles then also leads to removal of $NO_y$ from the stratosphere, known as denitrification. Denitrification slows the deactivation of active species since $NO_y$ would otherwise react e.g. to form $ClONO_2$, which does not photolytically destroy ozone (Crutzen and Arnold,

45     1986).

In some clouds Nitric Acid Trihydrate (NAT) is thought to nucleate on ice crystals (Höpfner et al., 2006), whilst in others it has been shown that crystalline NAX particles can form in conditions where ice is not thermodynamically stable (Mann et al., 2005;Tritscher et al., 2021). Whilst some authors have shown that relatively simple methods can recreate individual observations (Steiner et al., 2021), most have developed microphysical schemes to describe the nucleation process. A recent

review describes these efforts in detail (Tritscher et al., 2021). Current models of crystal formation mechanisms fall into three broad categories: those which explicitly treat heterogeneous nucleation of water ice and NAT (Hoyle et al., 2013;Engel et al., 2013;James et al., 2018), those which assume a heterogeneous nucleation mechanism, but model a constant nucleation rate per atmospheric volume (Carslaw et al., 2002), and those which assume nucleation occurs at the interface between the liquid droplet and the gas phase (Zhu et al., 2015). Nitric Acid Dihydrate (NAD) may also form, though this is not currently

considered in most atmospheric models (Grothe et al., 2008). Whilst meteoric material is often assumed to be the heterogeneous nucleus, terrestrial aerosol is also present in the stratosphere, also entrained in sulfuric acid droplets, but has been considered unlikely to contribute to nucleation in PSC as this tends to occur in descending airmasses originating from the mesosphere (Kremser et al., 2016). These descending airmasses contain considerably higher proportions of meteoric material compared to terrestrial aerosol, with up to 80 % of droplets in airmasses originating from above the stratosphere containing insoluble

inclusions (Weigel et al., 2014). Global models do not yet include a parameterisation of the nucleation process based on

laboratory measurements of reasonable heterogeneous nucleating surfaces. Where heterogeneous nucleation is treated explicitly, factors such as nucleating particle size and number concentration, and distributions of active sites have either been assumed or tuned to observed NAT particle concentrations (Grooß et al., 2014;Hoyle et al., 2013). However, models which are not constrained by a physical understanding, tested in laboratory experiments, may have a limited capacity for predicting future trends in ozone depletion.

Figure 1 shows possible pathways for meteoric material through the atmosphere, starting from various populations of incoming interplanetary dust. On heating by atmospheric friction the first processes that may occur are fragmentation or ablation of the incoming meteoroid, the solid core of the visible phenomenon known as a meteor (Carrillo-Sánchez et al., 2020;Subasinghe et al., 2016). Meteoroids which do not heat sufficiently to melt and ablate may still lose their more volatile carbon and sulfur components, resulting in sufficient weakening of the particle to allow fragmentation (Bones et al., 2022;Subasinghe et al., 2016). The remaining meteoric fragments would have a mineral and elemental composition similar to the incoming material except for these most volatile components (Taylor et al., 2012). Unablated meteoroids, partially melted cosmic spherules and meteoric fragments gravitationally sediment according to their size, whilst ablated metal atoms are oxidized to form a variety of species (oxides, hydroxides and carbonates) that then condense to form Meteoric Smoke Particles (MSPs) (Plane et al., 2015). MSPs are generally small enough that they do not gravitationally sediment at significant speeds, rather they are carried by the atmospheric circulation, with atmospheric lifetimes on the order of several years (Brooke et al., 2017;Dhomse et al., 2013). As MSPs and fragments descend through the atmosphere, gas-phase species can be taken up on their surfaces according to the gas phase abundance of the species (Frankland et al., 2015;James et al., 2017;Saunders et al., 2012). It has been shown that MSPs will become entrained within sulfuric acid droplets in the Junge layer (Brooke et al., 2017), and partially dissolve (Murphy et al., 2014;Bogdan et al., 2003), but the effect on meteoric fragments has not been previously considered.

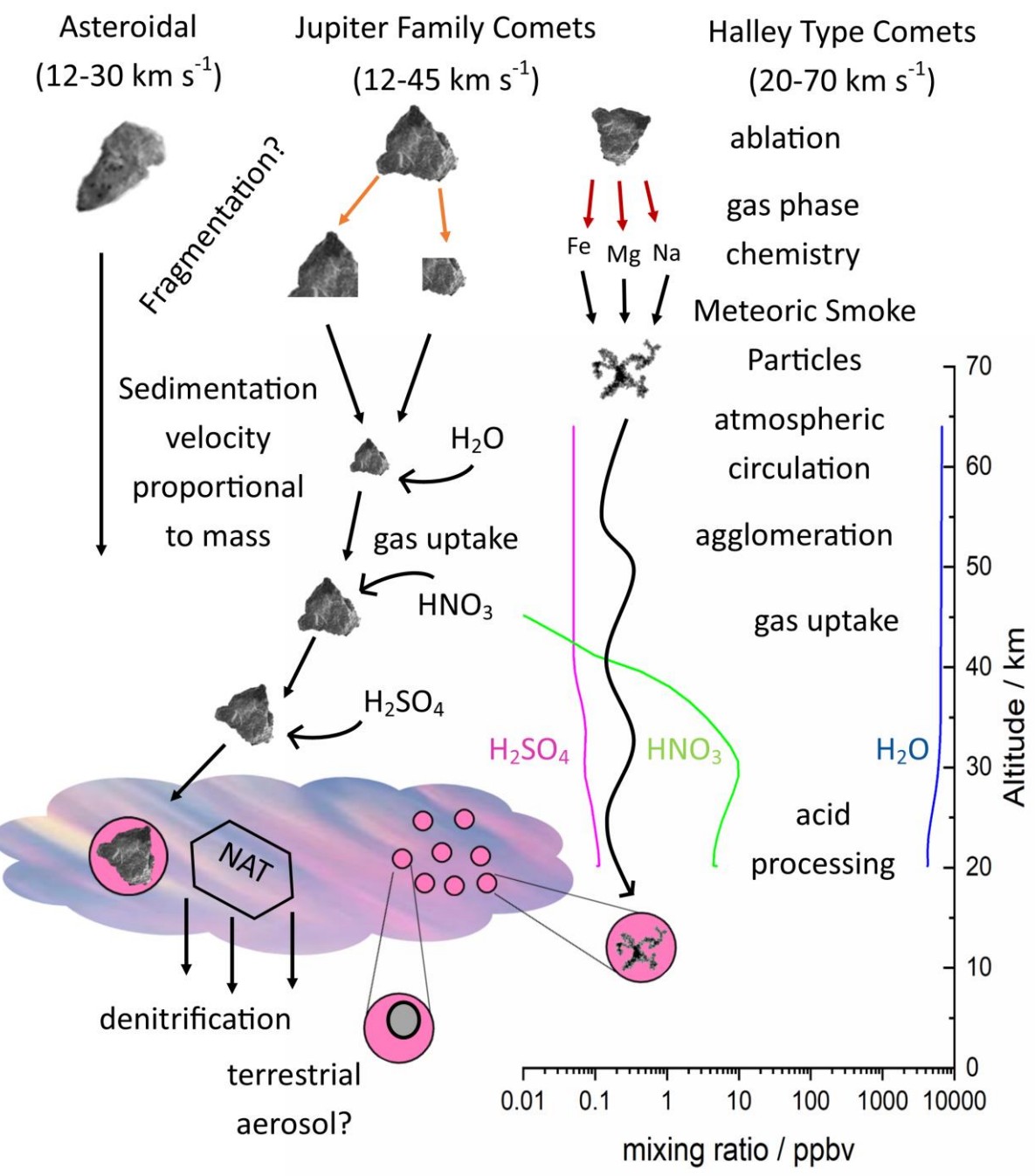

**Figure 1: Possible pathways (black arrows) for meteoric material through the atmosphere. Brackets show the limits of entry velocity. Types of aerosol and processes affecting them are shown. Altitudes and concentrations are indicative of approximately where each process is important. See text for detailed explanation.**

85

Relatively few laboratory studies have investigated the heterogeneous nucleation of NAX by meteoric material. Amorphous silica, used as an analogue for MSPs which have lost their metal content through acid leaching, was found to cause a heterogeneous effect in several studies (James et al., 2018;Bogdan et al., 2003). However the quantitative contribution of MSPs to atmospheric nucleation remains unclear, with different silica materials showing differing activity. Cosmic sprerules, which are a reasonable analogue for meteoric fragments, have also been shown to cause a heterogeneous effect (Biermann et al., 1996), and although this was not sufficient to explain clouds with high crystal number concentrations, it has been noted that this activity would be sufficient to explain nucleation in clouds with low crystal number concentration (Hoyle et al., 2013). Several ground meteorites covering a range of mineralogical compositions were tested as analogues for meteoric fragments, and found to have sufficient nucleation to explain an observed cloud crystal number density, however it was noted that these samples did not undergo any treatment to simulate either heating during fragmentation, or contact $H_2SO_4$, both atmospheric processes which could affect nucleation activity (James et al., 2018).

Heterogeneous nucleation can be controlled either by relatively rare active sites, or by a surface of relatively uniform activity (Murray et al., 2012). In either case, a solid nucleating particle (NP) included in the supercooled liquid droplet facilitates nucleation of the crystal (here we use "solid" to refer to the nucleating particle and "crystal" to the newly forming phase). With a rare active site, the activity of the NP is parameterised by the number of sites per unit area that cause nucleation under a specific set of conditions. In our previous work we used this method to parameterise the activity of a variety of analogues for meteoric material and compare them to an observed cloud (James et al., 2018). In previous modelling of PSC nucleation (Hoyle et al., 2013), it had been assumed that meteoric material causes heterogeneous nucleation kinetically at a rate determined by a distribution of active sites according to the Classical Nucleation Theory (CNT). In CNT a rate of nucleation is calculated based on the system's ability to overcome barriers to diffusion to the interface between the liquid and the cluster of nucleating crystal, $\Delta F$, and to formation of the crystalline molecular cluster of critical size, above which the crystalline phase grows freely, $\Delta G^*$. This rate, $J_{het}$, is defined as equation E1 (Pruppacher and Klett, 1978;Murray et al., 2012;Fletcher, 1958).

$$J_{het}/cm^{-2}s^{-1} = \frac{n_{mol}k_BT}{h}e^{-\Delta F/k_BT}e^{-\Delta G^*f_{het}/k_BT} \tag{E1}$$

where $n_{mol}$ is the number of molecules per unit volume in the liquid and $k_B$ and $h$ are the Boltzmann and Planck constants, respectively. The geometric factor, $f_{het}$, can be conceptualised as the relative reduction in the crystalline cluster volume required for critical size. Most generally, it is defined by E2.

$$f_{het} = \frac{1}{2}\left(1 + \left(\frac{1-mX}{\varphi}\right)^3 + X^3\left(2 - \left(\frac{3(X-m)}{\varphi}\right) + \left(\frac{(X-m)}{\varphi}\right)^3\right) + 3mX^2\left(\frac{(X-m)}{\varphi} - 1\right)\right) \tag{E2}$$

where $X = \frac{r_{NP}}{r^*}$ is the ratio between the radius of the nucleating particle, $r_{NP}$, and the critical cluster, $r^*$, the contact parameter $m = \cos\theta$ where $\theta$ is the contact angle, and $\varphi = \sqrt{(1 + X^2 - 2Xm)}$. The radius of the critical cluster size can be determined from thermodynamic properties of the system by E3.

$$r^* = \frac{2V_{mol}^2\sigma}{(k_B T \ln(S))^2} \tag{E3}$$

where $V_{mol}$ is the volume of one molecule of solid composition, $S$ is the saturation ratio (the ratio of free energies of the supercooled liquid to the system at thermodynamic equilibrium), and $\sigma$ is the interfacial energy between the crystal and liquid. $\Delta G^*$ can similarly be determined from thermodynamic properties:

$$\Delta G^* = \frac{16\pi}{3}\frac{V_{mol}^2\sigma^3}{(k_B T \ln(S))^2} \tag{E4}$$

Following nucleation the crystalline phase will grow at the expense of the liquid droplet, and since the vapour pressure over the solid phase is lower than that over the liquid phase, the composition of other droplets can also be affected (Carslaw et al., 2002). The result can be a cloud where relatively few crystals form, grow rapidly and sediment until they reach a region warm enough for the particles to evaporate (Voigt et al., 2005), causing a vertical redistribution of $NO_y$, including nitric acid (Fueglistaler et al., 2002). At altitudes which are denitrified, this can lead to enhanced ozone depletion because a reduction of $NO_x$ species slows deactivation of Cl and Br catalytic ozone destroyers. Ozone depletion is therefore sensitive to the nucleation of crystalline PSCs, both through the available catalytic aerosol surface and through the denitrification process (Wegner et al., 2012).

The activity of some heterogeneously nucleating materials has been found in the past to be affected by the presence in solution of, or pre-treatment by, acids or other solutes (Whale et al., 2018;Wex et al., 2014). In particular $H_2SO_4$ has been found to deactivate many ice nucleating materials, even where some other acids had little effect (Kumar et al., 2019;Fahy et al., 2022). To date no systematic test has been made on the effect of $H_2SO_4$ on heterogeneous nucleation in PSCs, although there is $H_2SO_4$ present in the liquid stratospheric aerosol from which crystalline PSC particles form (Carslaw et al., 1997).

In this study we assess the potential pathways to the heterogeneous nucleation of NAT through nucleation on meteoric fragments and nucleation on MSPs. In the first part of the paper we experimentally explore the sensitivity of the nucleation activity of meteoric fragments to $H_2SO_4$ and the heating that occurs on entry to the atmosphere. We then use sedimentation modelling and comparison to measurements of meteoric input to the Earth to assess the likelihood that meteoric fragments contribute to crystal nucleation in PSCs and constrain their flux. In the second part we theoretically assess whether nucleation on silica particles resulting from acid processing of MSPs might contribute to the NAT population. In the past it has been shown that silica can nucleate NAT, but its activity varies massively (James et al. 2018; Bogdan et al., 2003), with smaller silica particles nucleating NAT less effectively. We develop a size dependent nucleation parameterisation for nucleation of

NAT on silica, constrained by our previous experimental results, to assess the likelihood that acid processed MSPs contribute to crystal nucleation in PSCs. We investigate specifically whether nucleation activity of these heterogeneous materials is sufficient to explain observed cloud crystal number concentrations in the absence of water ice.

## 2 Methods

To determine the heterogeneous activity of analogues for meteoric material, arrays of 1 µl droplets were cooled until they crystallised and the nucleation events observed using a liquid nitrogen-cooled cold stage shown in Figure 2 a, described previously (Holden et al., 2019). Nucleation, crystal growth and melting were observed using a CMEX-5 pro CCD camera (Figure 2 b-f). Temperature was measured using a platinum resistance thermometer and controlled by balancing a constant liquid nitrogen cryogen flow with resistive heating cartridges embedded in the aluminium cold stage. Droplet arrays were pipetted onto a hydrophobic glass slide, sealed into a cell by surrounding with a viton O-ring used as a spacer and covered with a second glass slide, then a dry nitrogen flow was used to prevent icing of the upper surface of the glass slide during cooling. Reflected illumination was provided by four plane polarised LEDs.

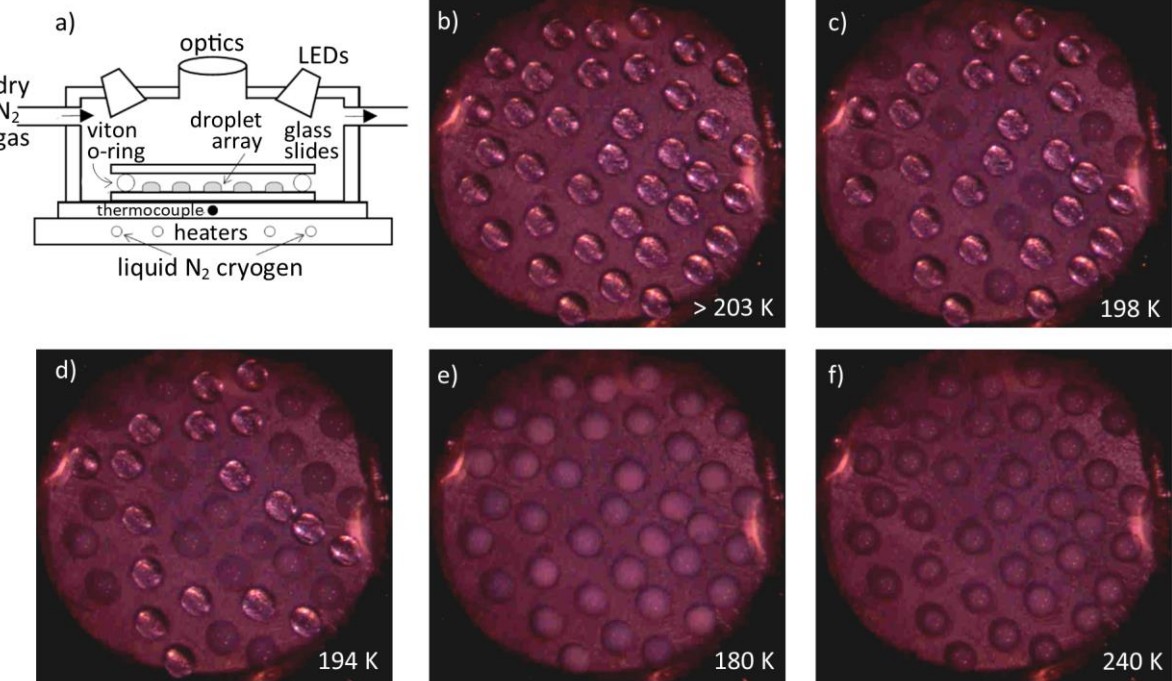

Figure 2: a) Diagram showing experimental apparatus. An aluminium stage is cooled by a constant flow of liquid nitrogen and temperature and rate of change is controlled by heating cartridges embedded in the aluminium. A hydrophobic glass slide is placed on this surface, an array of 1 µl droplets is pipetted onto the slide, surrounded by a greased viton o-ring and covered with a second

slide to minimise concentration change by mass transfer between liquid and vapour. A dry nitrogen flow is passed over this sealed cell to prevent ice deposition from the atmosphere at low temperature. The cell is lit using plane polarised LEDs and reflected light images recorded with a microscope equipped with a CCD camera. b-e) droplet arrays showing nucleation and crystal growth on cooling, f) partially melted array on warming.

An analogue for meteoric fragments was produced by grinding a sample of the Allende meteorite (Clarke et al., 1971) with a pestle and mortar until it passed through an 18 μm diameter sieve (Endecotts test sieves), referred to as 'recently ground' to distinguish from samples held (to these experiments performed in 2020) since our previous work in 2016 (James et al., 2018). Ground meteorites are considered good analogues for meteoric fragments since they have similar mineralogical and elemental composition to the bulk of incoming material. Some of this recently ground material was further treated to simulate frictional heating during atmospheric entry, which leads to meteor fragmentation, by annealing (heating) under an $N_2$ atmosphere. Temperature was ramped at 16 K min$^{-1}$ to 700 K in a tube furnace (Carbolite Gero) and held for half an hour. Under this heating regime, essentially all the refractory organic material in the ground meteorite pyrolyzes (Bones et al., 2022). This Annealed Allende (AA) sample was stored in a desiccator before samples were removed to make up droplet suspensions. All nucleating material samples were analysed to determine their BET surface area. Aqueous acid 'control' solutions and suspensions of heterogeneous material were prepared by mass dilution from 69 wt% $HNO_3$ (Aristar, trace analysis grade), and 95 wt% $H_2SO_4$ (Acros organics, 96 %, other $H_2SO_4$ suppliers and grades were tested but found to have higher nucleation temperatures in control experiments without added heterogeneous material i.e. higher levels of nucleation active contamination). Suspensions were made up by first preparing the acid solution in a cold sonic bath (20 % aqueous propylene glycol cooled with $CO_2$ cardice to below 273 K, ranging down to 250 K), then adding the appropriate amount of meteoritic fragment analogue and sonicating for 10 minutes. Solutions were cooled to limit the possible chemical alteration of the heterogeneous nucleating material by acid solution, which might be faster at room temperature than under stratospheric conditions. Suspensions were stored in the cold bath and a new array of droplets tested at approximately hourly intervals to check for any time dependence of acid sensitivity.

Droplet arrays containing a range of Allende meteorite concentrations were prepared, either with or without 0.5 wt% $H_2SO_4$. This $H_2SO_4$ concentration was chosen as it is similar to the lowest at thermodynamic equilibrium at temperatures where NAX might form in the absence of water ice (Carslaw et al., 1997). For example, in an atmosphere containing 5 ppmV $H_2O$, 15 ppbV $HNO_3$ and 0.5 ppbV, $H_2SO_4$ concentration in the liquid phase at equilibrium would reach 0.5 wt% at 191.2 K, where saturation with respect to water ice is 0.94 (Clegg et al., 1998). In some cases the $HNO_3$ concentration was varied from 40 up to 43 wt%. This reduces the $H_2O$ ice melting point by up to 20 K, leading to a greater proportion of the nucleation occurring above the ice melting point and less ambiguity in the phase which initially nucleates (Clegg et al., 1998). A full list of samples tested is shown alongside a summary of the results in Table 1 (see Section 3, below).

In an attempt to test the effect of $H_2SO_4$ on nucleation by analogues for MSPs, ternary solutions containing fumed silica were also tested, and found to have nucleation temperatures below the instrument background. However at room temperature these

suspensions were found to form a gel within several hours. Given the temperature (>250 K) at which the suspensions were made up, which may allow gel formation through increased dissolution at the particle surface, this suggests that gel formation may be an artefact of our laboratory conditions and not be important in the atmosphere. Hence, gel formation precludes experiments on fumed silica in solutions containing sulfuric acid. Atmospheric droplets are known to contain solid silica particles, since the variable hit rate of these particles by the laser ionisation sources of atmospheric aerosol mass spectrometers leads to signal broadening (Murphy et al., 2014). This is discussed further in Section 4.4, below.

Arrays of droplets were pipetted at 283 K, above the dewpoint in the room but again minimising the temperature, covered and cooled at 5 K min$^{-1}$ to 210 K, then 1 K min$^{-1}$ until all droplets were frozen or to a minimum temperature of 150 K. This is approximately the glass temperature of HNO$_3$ aqueous solutions, so further crystallisation below this temperature is unlikely (Frey et al., 2013). In several cases adjacent droplets coalesced either due to vibrations or during crystal growth, these events were discarded from the data set. Samples were then warmed at 5 K min$^{-1}$ to 283 K, with the melting points recorded. Observed melting began at 231 ± 1 K, corresponding to the NAT / H$_2$O ice eutectic temperature and ended at temperatures within 1 K of the NAT melting point ($S_{NAT}$ = 1.01 ± 0.08) for the solution concentration applied (Clegg et al., 1998). This variability could be the result of two effects, either poor measurement and control of temperature, or a poor seal of the sample cell, leading to a change in the droplet concentration over the experimental time-period. If the discrepancy from the expected melting point were entirely due to a change in concentration, and the measured melting temperature were taken as a measure of the final HNO$_3$ concentration, the change would be 0.5 ± 0.9 wt%. Note that the greater vapour pressure of H$_2$O compared to HNO$_3$ would suggest that any improper seal should lead to a concentration and therefore melting point increase. We therefore consider the variability in melting point to be mainly a result of uncertainty in the measurement and control of temperature, with the stated variability in concentration representing an upper limit. In a few control experiments changes in brightness occurred on warming between 205 (the NAD / H$_2$O ice eutectic) and 230 K. These could be indications of metastable phases either melting or recrystallizing to stable phases that are not represented in the currently accepted H$_2$O / HNO$_3$ phase diagram. We did not investigate this further since the primary goal was to quantify nucleation of crystalline phases.

**3 Effect of H$_2$SO$_4$ and heating on nucleation activity of meteoric fragments**

Observed fraction frozen data are shown in Figure 3, for samples with and without sulfuric acid. Control experiments of binary H$_2$O / HNO$_3$ solutions are in good agreement or show a lower nucleation temperature when compared with our previous experiments using a Stirling engine cold stage (James et al., 2018). We note that the heterogeneous nucleation temperatures for one sample decreased with the time meteoric particles were suspended in HNO$_3$ at temperatures from 250-270 K prior to the freezing assay, gradually falling to the baseline in approximately four hours, whilst for most experiments any changes in activity were less than the variability between repeat experiments (e.g. backgrounds or similar first runs). Addition of H$_2$SO$_4$

increases the background nucleation temperature and decreases the heterogeneous nucleation temperatures for most samples. The result is that many runs show no nucleation activity above the instrument baseline. Similar deactivation of nucleation activity by acid treatment has been observed for the water ice system with effects such as leaching of surface cations,

dissolution of surface mineral active sites and deposition of saturated salts proposed as explanations (Fahy et al., 2022). The overriding picture here is one of variability, it is likely that the lack of reproducible behaviour between samples of meteorites reflects a difference in the phases within the heterogeneous meteorite which control nucleation (Taylor et al., 2012). A notable exception to the loss of activity on inclusion of $H_2SO_4$ is the annealed Allende sample, where at least the most active third of droplets nucleate significantly above the baseline in all repeat experiments.

The increased nucleation temperatures observed in control experiments with addition of $H_2SO_4$ are most likely due to impurities in the $H_2SO_4$. Wise et al. (2003) observed a similar temperature increase on addition of metal salts to $H_2SO_4$ solutions; however, we did not observe a similar effect in binary $HNO_3$ / $H_2O$ solution (James et al., 2018). Since we tried several $H_2SO_4$ brands and grades and found a variable increase in nucleation temperature, we consider it unlikely that the increased crystallisation temperature on addition of metal salts is likely to occur in the atmosphere. Nucleation on

heterogeneous impurities is also consistent with the variability in the background nucleation temperatures, which is significantly greater for experiments containing $H_2SO_4$ than those with only $H_2O$ and $HNO_3$.

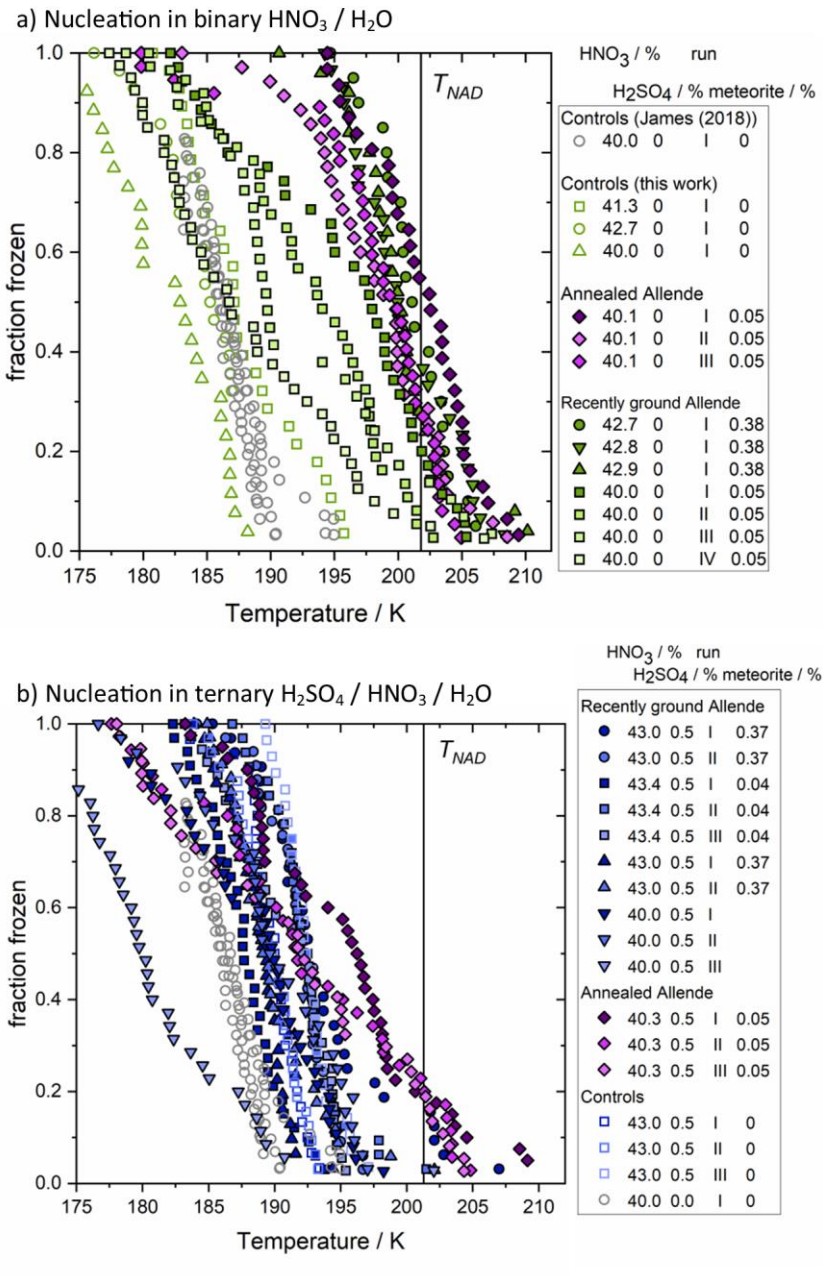

**Figure 3: Fraction frozen results for meteoric fragment analogues a) binary solution and b) ternary. Symbol shape varies for each experiment and shading for repeats, which were at intervals of approximately one hour with the sample suspension stored at <270 K in between. Open symbols show control experiments, grey open circles are data from James et al. (2018). Legend shows the concentration of acid species and nucleating particles, and the repeat status for each data set. The melting temperature of NAD in a 40 wt% $HNO_3$ solution is indicated by a vertical line. The data are summarised in Table 1. Uncertainty in temperature is 1 K, see Methods Section.**

**Table 1: Summary of samples prepared (see Methods section) and observed heterogeneous activity**

| | BET surface area / $m^2$ $g^{-1}$ | $H_2O$ / $HNO_3$ = 60/40 | $H_2O/HNO_3/H_2SO_4$ = 59.5/40/0.5 |
|---|---|---|---|
| Control (no added meteoric material analogue) | 0 | Nucleation temperatures agree with James et al. (2018). | Nucleation warmer and more variable than binary solutions. |
| Allende sample from James et al. (2018) | $1.22 \pm 0.15$ | Nucleation temperatures agree with James et al. (2018). | Nucleation cooler than binary solutions, within baseline range. *i.e.* deactivated. |
| Recently ground Allende | $5.80 \pm 0.05$ | Initially active but loses activity in $HNO_3$ over several hours of immersion in HNO3 solution at 250-270 K. | Deactivated. |
| Annealed Allende | $1.78 \pm 0.56$ | Slightly more active than other meteoric fragment analogues either here or in James et al. (2018). | Activity above baseline and in good agreement with James et al. (2018) in 40-60 % of droplets, resistant to acid over several hours. |
| Annealed Allende stored in a desiccator for one week | $1.78 \pm 0.56$ | Deactivated | Deactivated |

Even focussing on debris from a single meteorite fall, significant variability in heterogeneous nucleation behaviour is observed between very similarly treated samples. This is reasonable given the known heterogeneity of meteorite minerology. Such heterogeneity is also thought to be present in micrometeorites (Taylor et al., 2012). Annealing to simulate atmospheric entry

and fragmentation seems to reduce the samples' sensitivity to acid exposure, but a reduction of activity in dry room temperature air was still observed. Since atmospheric lifetime with respect to gravitational settling is related to particle size and mass, this may constrain a minimum size below which a meteoric fragment's atmospheric lifetime is long enough to permit extensive acid processing and a reduction in nucleation activity. However, the difference between conditions such as temperature,

pressure and relative humidity in our laboratory experiment and the upper atmosphere is significant, so we do not recommend any quantitative constraint on the particle size that could be deactivated by acid processing.

To facilitate a comparison with our previous experiments, and determination of the atmospheric relevance of our observed nucleation efficiencies, Figure 4 presents these results as $n_s$:

$$n_s = -\frac{\ln(1-f(S_x))}{s} \qquad\qquad\qquad\qquad (E5)$$

the number of active sites per unit surface area of solid inclusion active at a given saturation ratio, where $f(S_x)$ is the saturation dependent fraction of droplets which have crystallised and $s$ is the surface area of the nucleating material which was calculated from the mass concentration of heterogeneous material and the BET surface area (see Table 1) (Murray et al., 2012). Here experiments containing $H_2SO_4$ and using the recently ground Allende meteorite are shown as transparent symbols, since their nucleation temperatures were not significantly above the experimental baseline, and should therefore be considered an upper limit to the activity of the samples. Uncertainty in this data comes from several sources; variability in temperature control, the measured BET surface area, and the distribution of active sites through the droplet population. Particles that carry nucleating active sites are distributed randomly between droplets and this variability represents a source of uncertainty. As an example, one data set is presented with error bars (others follow the same trend, but are excluded for clarity) that show the contribution of the randomness of active site distributions, calculated using a Monte-Carlo method as described in (Sanchez-Marroquin et al., 2020). The first few and last few points in each set of data carry the largest uncertainty. However, other uncertainties, such as the Poisson counting error and droplet temperature, dominate for most data points as can be seen in the variability between repeat runs. For example, the repeat runs of the Allende samples with no $H_2SO_4$ (green hexagons and leftward facing triangles in Figure 4) are generally within a factor of ~2 of one another in $n_s$, or within 3 K of one another in temperature (green circles and triangles in Figure 3). Given melting point measurements indicate an uncertainty in temperature of 1 K, uncertainties from temperature measurement, distribution of active sites and material through the droplets, and Poisson counting statistics likely contribute to the uncertainty in the quoted $n_s(T)$.

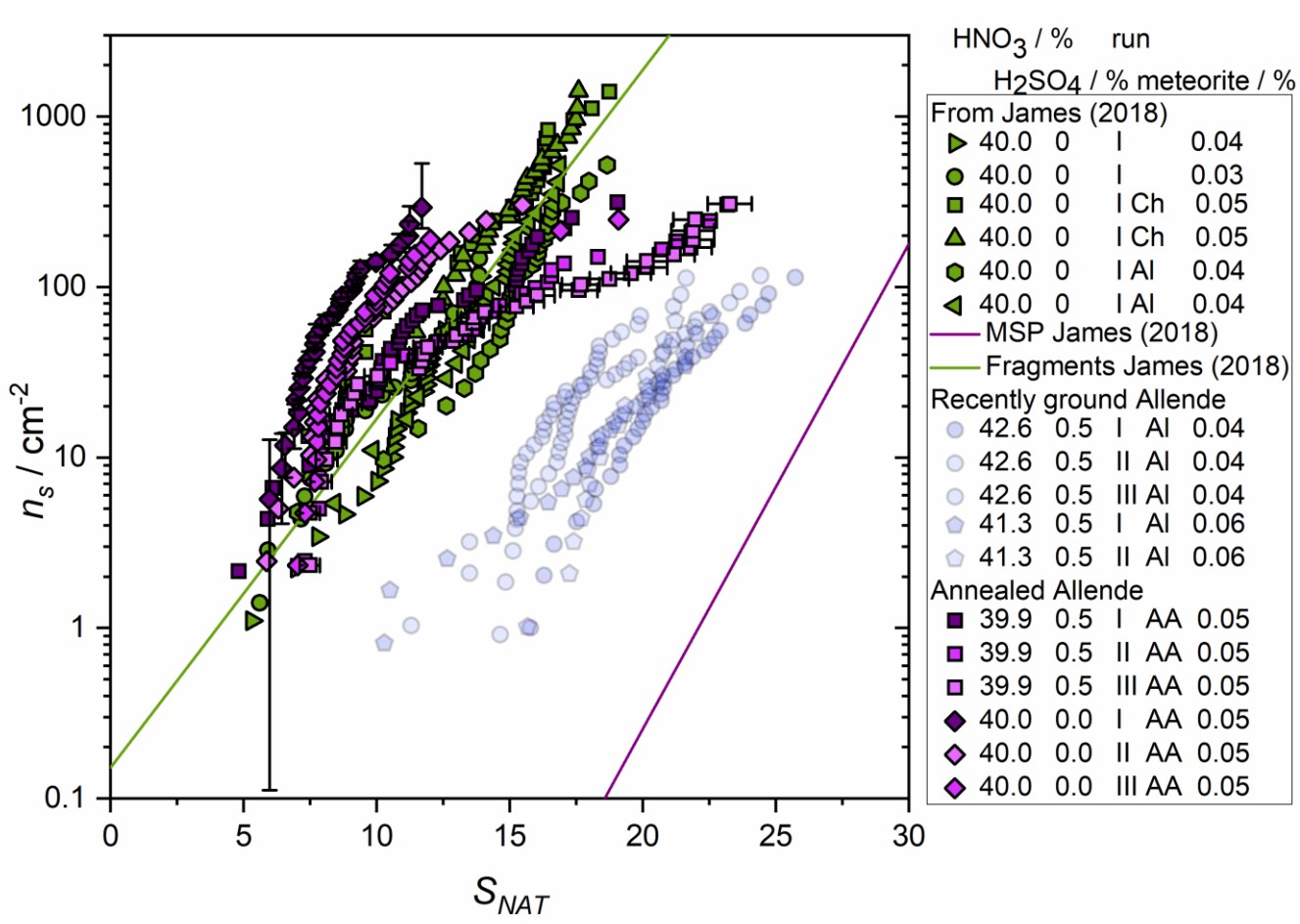

**Figure 4: Nucleation effectiveness represented by the number of sites active per unit surface area, $n_s$, under particular conditions, here represented by the saturation ratio with respect to Nitric Acid Trihydrate (NAT) of meteoric fragment analogues with (blue) and without (green) $H_2SO_4$. As in James et al. (2018), unlabelled data is from experiments using the North West Africa 2502 meteorite, Ch indicates the Chergach meteorite, Al indicates Allende, see that work for details of sample preparation and experimental setup. Here AA (purple data) indicates a sample of the Allende meteorite annealed to simulate relatively mild heating on atmospheric entry which could lead to fragmentation (see text for details). Green symbols with the exception of diamonds are from James et al. (2018). Blue data shows experiments containing $H_2SO_4$. One data set is presented with vertical error bars which illustrate the contribution to the uncertainty of the randomness of the distribution of nucleating sites between droplets. A second data set shows the temperature uncertainty of the instrument propagated into saturation terms. These uncertainties are of similar magnitude for other data sets but are omitted for clarity.**

These experimental results demonstrate that, whilst acid suspension can have complex effects on the nucleating activity of fragmented meteoric material, at least some subset of the material can maintain activity, in agreement with our previous parameterisation (James et al., 2018). This allows us to carry out a more thorough evaluation of whether sufficient fragments are supplied to the lower stratosphere, with reference to the latest understanding of meteoritics and atmospheric entry. To do

this, we varied the available surface area per droplet of nucleating particles and evaluated the final concentration of crystals in a 1-D atmospheric model of nucleation, compared to observations. This model uses calculated back trajectory temperatures and measured atmospheric $HNO_3$ concentrations to derive $S_{NAT}$, and parameterised nucleation activity to derive the number density of crystallised particles, $N_{NAT}$, and compares these to atmospheric observations (Voigt et al., 2005). Processes such as

growth and sedimentation of crystalline particles out of the air mass are not considered. Using the same model in our previous work we showed that with this parameterisation of fragment activity, an atmospheric surface area density of 0.76 $\mu m^2$ $cm^{-3}$ produced around $2 \times 10^{-5}$ NAT crystals $cm^{-3}$. We estimate 0.25 $\mu m^2$ $cm^{-3}$ as a minimum surface area density that could produce the lowest observed crystal concentration of $6 \times 10^{-6}$ $cm^{-3}$ (see Figure 11, below).

Assuming that the mass of meteoric fragments is such that their transport is completely dominated by gravitational

sedimentation, we can combine the fall speed of a given particle with constraint of the total input of meteoric material to the Earth to estimate whether this surface area density might be available.

To calculate fall speed we assume a mass density of 2.2 g $cm^{-3}$ for meteoric fragments and sedimentation velocity calculated using Stokes' law for a spherical particle falling through a stationary fluid (Jacobson, 2005). Figure 5 shows the resulting atmospheric lifetime of meteoric fragments with respect to gravitational sedimentation. For example, particles of around

0.5 $\mu m$ radius or larger will fall into the stratosphere within one day. Horizontal atmospheric winds on the order of 10s m $s^{-1}$ will move the particles on the order several km in that time, meaning that there will not be significant redistribution of fragments toward the mesospheric winter pole.

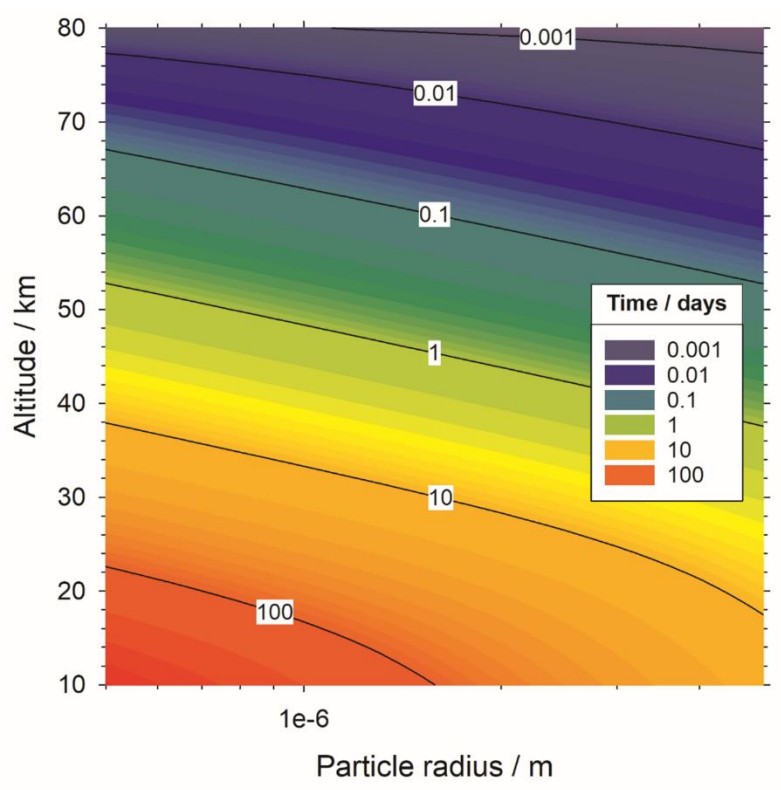

**Figure 5: Cumulative time for a meteoric particle of given size to sediment gravitationally to a given altitude.**

We can then use known constraints of the Meteoric Input Function (MIF, ton day[-1]) to estimate whether fragments of a given size could significantly influence nucleation in the polar stratosphere. The most recent investigations of the Earth's MIF suggest that each day 8.3 tons of meteoric material ablate and ultimately produce MSPs, 5.5 tons partially melt to produce cosmic spherules such as those observed in the South Polar Water Well (Taylor et al., 2012), and a final 14.2 tons do not heat sufficiently to melt (Carrillo-Sánchez et al., 2020). Any particles entering the Earth's atmosphere but not included in this would have to provide minimal contribution to the zodiacal scattered light, fragment to sizes smaller than 50 µm radius and provide negligible Na and Fe input to the mesosphere. Observations by aircraft in the polar stratosphere have found significant numbers of micron-size particles, with MIF estimates from 77 to 375,000 tons day[-1] required to explain the number of particles collected (Weigel et al., 2014). Single particle mass spectrometer measurements on aircraft also detect significant numbers of meteoric particles (Schneider et al., 2021;Murphy et al., 2021;Adachi et al., 2022), and are able to investigate their composition and atmospheric distribution, but since the sulfate components are also present in stratospheric aerosol in variable amounts it is not trivial to derive MIF information from these measurements. A reasonable upper limit to the MIF comes from the Long Duration Exposure Facility (LDEF), which measured pits on surfaces exposed in near-Earth orbit and an assumed velocity of incoming particles to estimate a mass flux of $110 \pm 55$ ton day[-1] (Love and Brownlee, 1993). Accounting for the mass required to explain

ablated metals measured in the atmosphere and particles present in the south polar water well collections, we derive an upper

limit for the mass of incoming material which could fragment at 137 ton day$^{-1}$.

Assuming for simplicity that this incoming mass fragments to a monodisperse particle size and is transported only by sedimentation we can calculate a resulting surface area density at PSC altitudes. This is shown in Figure 6. The implication is that all of the incoming particles would have to fragment to radii of <100 nm to provide a significant contribution to observed crystalline PSC.

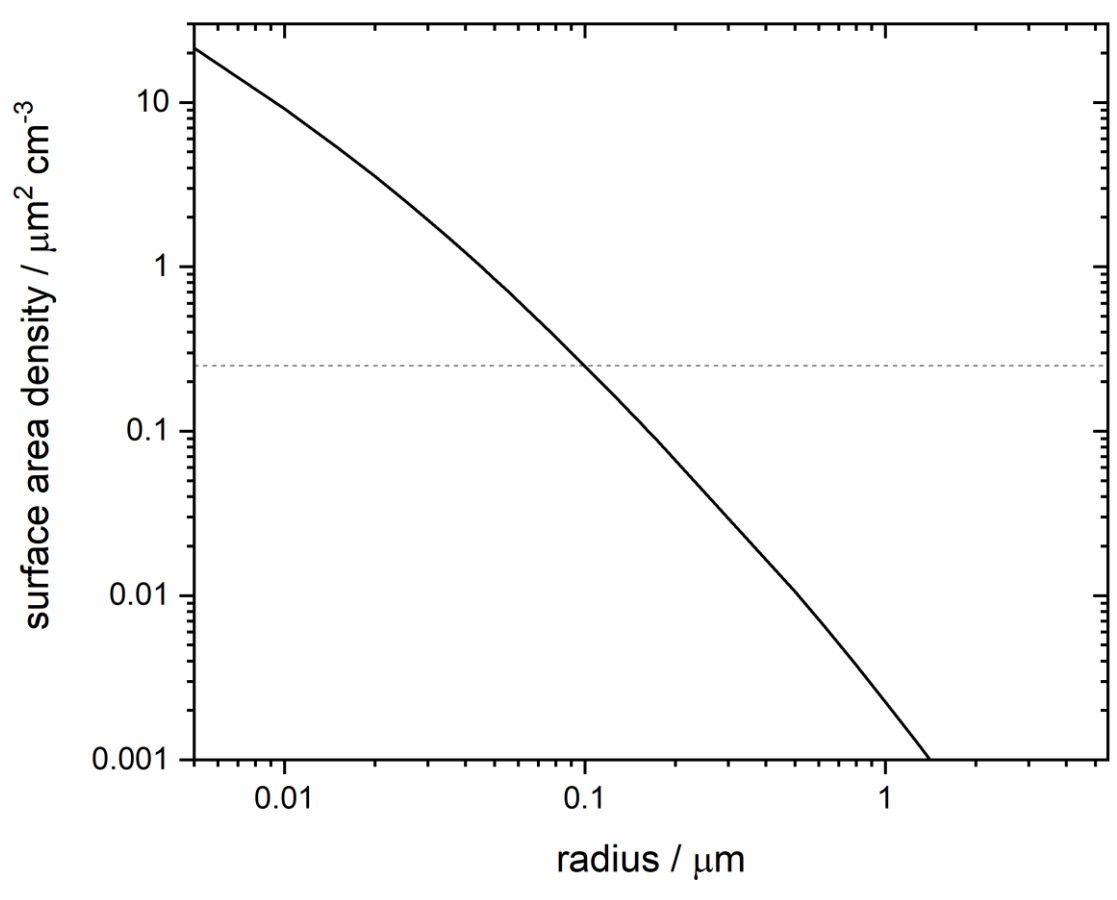

**Figure 6: Surface area density of meteoric fragments assuming that a reasonable upper limit of 137 ton day$^{-1}$ fragment to the radius shown and sediment gravitationally to 20 km. Dashed horizontal line shows the 0.25 µm$^2$ cm$^{-3}$ required to produce the lower limit of observed crystal numbers in water ice free PSCs.**

Recent laboratory studies have suggested that meteorites heated to simulate the fragmentation process often become stronger (require more pressure for an atomic force microscope tip to break the surface) if somewhat more brittle (tip penetrates deeper once the surface is broken) (Bones et al., 2022). Recent stress testing of material from comet 67P suggests that the particles are made up of agglomerated fractals with highly non-spherical primary particle size of at least several hundred nm equivalent radius (Mannel et al., 2019). These studies do suggest a sub-set of loosely agglomerated, relatively weak cosmic dust which may fragment, though constraining the influx of such a material precisely enough for a quantitative comparison between nucleation by MSPs and fragments is not currently possible.

Based on these various analyses, some statements can be made about the potential for fragments to act as competitive heterogeneous nucleating particles in PSC. Firstly, micron-sized fragments, whilst they do reside in the lower stratosphere for time periods similar to the cloud lifetime, would require a very large meteoric flux to compete as nucleating particles. The upper end of estimates based on aircraft observations of large particles might allow for such high fluxes (Weigel et al., 2014), but they would be extremely difficult to reconcile with observations and modelling of dust in our solar system. Secondly, if observations from comet 67P can be generalised to all cosmic dust (Mannel et al., 2019), and the minimum fragment size is some hundreds of nm equivalent radius, a reasonable upper limit to the MIF is unlikely to be sufficient to contribute to crystallisation. Fragmentation to 100 nm radius would be required for our upper limit mass flux to produce our lower limit fragment surface area density, and this is somewhat smaller than the primary particle size observed from comet 67P. Finally, fragmentation to 10s nm radius would be required to agree with MIF estimates of unablated material consistent with the zodiacal light, cosmic spherule collections and mesospheric metal fluxes to cause competitive nucleation (Carrillo-Sánchez et al., 2020). At these sizes particles would be carried by atmospheric circulation, concentrated in the mesosphere towards the winter pole and partially dissolve in acid droplets, much like MSPs. However, mechanical break up of particles to such small sizes is not commonly possible even under controlled laboratory conditions (Wang and Forssberg, 2006).

The micron-size particles observed by aircraft in the stratosphere, some of which have compositions suggesting an extra-terrestrial origin (Ebert et al., 2016), could be meteoric fragments. If the 0.1 ppbm of refractory (stable under an electron beam) particles observed in that study were all meteoric fragments of 0.5 µm radius they would have surface area density of 0.03 $\mu m^2 cm^{-3}$ and require a mass flux of 270 tons day$^{-1}$, however not all of these particles are meteoric. Balloon-borne collections of refractory particles have shown a factor of 5 less refractory particles (Deshler et al., 2003). Atmospheric modelling of the distribution of fragmented material of reasonable size would assist with determining whether these particles could be meteoric fragments, but more information about the incoming material and fragmentation process would still be required to reach a firm conclusion. Taking all of this into account, we conclude that it is unlikely that meteoric fragments contribute as NAT nucleating particles in PSCs.

## 4 Size dependent nucleation by meteoric smoke particles

We now explore whether MSPs in reasonable atmospheric concentrations and size distributions would have sufficient activity to explain observed cloud crystal number densities. Silica has been shown to nucleate NAT in the past (James et al., 2018; Bogdan et al., 2003). James et al. (2018) showed that fumed silica particles of around 6 nm nucleated NAT much less effectively than micron scaled particles of silica. It is well-known that the nucleating ability of a particular material decreases dramatically when the grain size approaches the size of the critical cluster, typically < 10 nm (Pruppacher and Klett,

1978;Fletcher, 1958). This is because the heterogeneous nucleus effectively forms some of the volume of the critical cluster, reducing the barrier to critical cluster formation, and a small heterogeneous nucleus is not able to significantly reduce the volume of the critical cluster. Given silica particles in the stratosphere have a size distribution with a mean size in the 10s of nanometers (Bigg, 2012), it seems reasonable that some of these particles might nucleate NAT in the stratosphere. In order to explore this possibility, we construct a nucleating particle (NP) size-dependent model using CNT (Pruppacher and Klett,

1978;Fletcher, 1958), and use our data on heterogeneous nucleation by MSP analogues with available thermodynamic data to constrain this model. We then combine this with atmospheric measurements of the size distribution of available material and back-trajectory temperature and $NO_y$ profiles as in our previous work (James et al., 2018), thereby predicting the cloud crystal number density which can be compared with observations.

    Equations E1-4 provide a kinetic framework to determine nucleation rates from thermodynamic quantities and the empirically

determined contact angle, which relates to the activity of the nucleating particle surface. The saturation of the system is the fundamental variable, which leads the nucleation to be affected by the droplet environment. Whilst it is not clear that each of these quantities is independent of the external conditions (Knopf et al., 2002), this is a common assumption (Koop and Murray, 2016). In the case of the nitric acid / water system, these thermodynamic quantities are not well established, but experimental observations do exist which allow their values to be constrained. The most important of these are the surface tension, $\sigma$, and

the diffusion barrier, $\Delta F$. Since the phase which is observed growing and melting is not necessarily that which first nucleates we cannot rule out the nucleation of metastable phases such as $\alpha$-NAT or in many cases either polymorph of NAD (Weiss et al., 2016;Wagner et al., 2005a). Note here that because we observe significant nucleation above the NAD melting point, we assume that a NAT phase is nucleating, and that it has the thermodynamic properties observed for NAT nucleation reported in the literature (i.e. it is the same polymorph as was observed in those other experimental studies). We go on to show that this

assumption provides an internally consistent explanation of a range of experimental data, indicating that it is reasonable.

### 4.1 Diffusion activation energies

    The diffusion activation energy has been measured in stoichiometric (*i.e.* 3:1 $H_2O$:$HNO_3$) solution (Tisdale et al., 1997), and found to vary over 48.5-36.8 kJ mol$^{-1}$ at 185-200 K. However, here we were able to determine a value at a more atmospherically relevant liquid concentration by using the temperature-dependent crystal grown rate in our experiments. The growth rate of

the advancing crystal / liquid interface between video frames in cm s$^{-1}$ was measured for crystals forming in droplets of 40 wt%

HNO$_3$. An Arrhenius fit to these crystal growth rates is shown in Figure 7, resulting in a value measured between 182 and

207 K of 35.3 ± 4.3 kJ mol$^{-1}$, in agreement within error with the previous measurement. We note that both our and the literature

value are somewhat higher than the 25 kJ mol$^{-1}$ derived from measurements of HNO$_3$ diffusion coefficients on ice (Luo et al.,

2003).

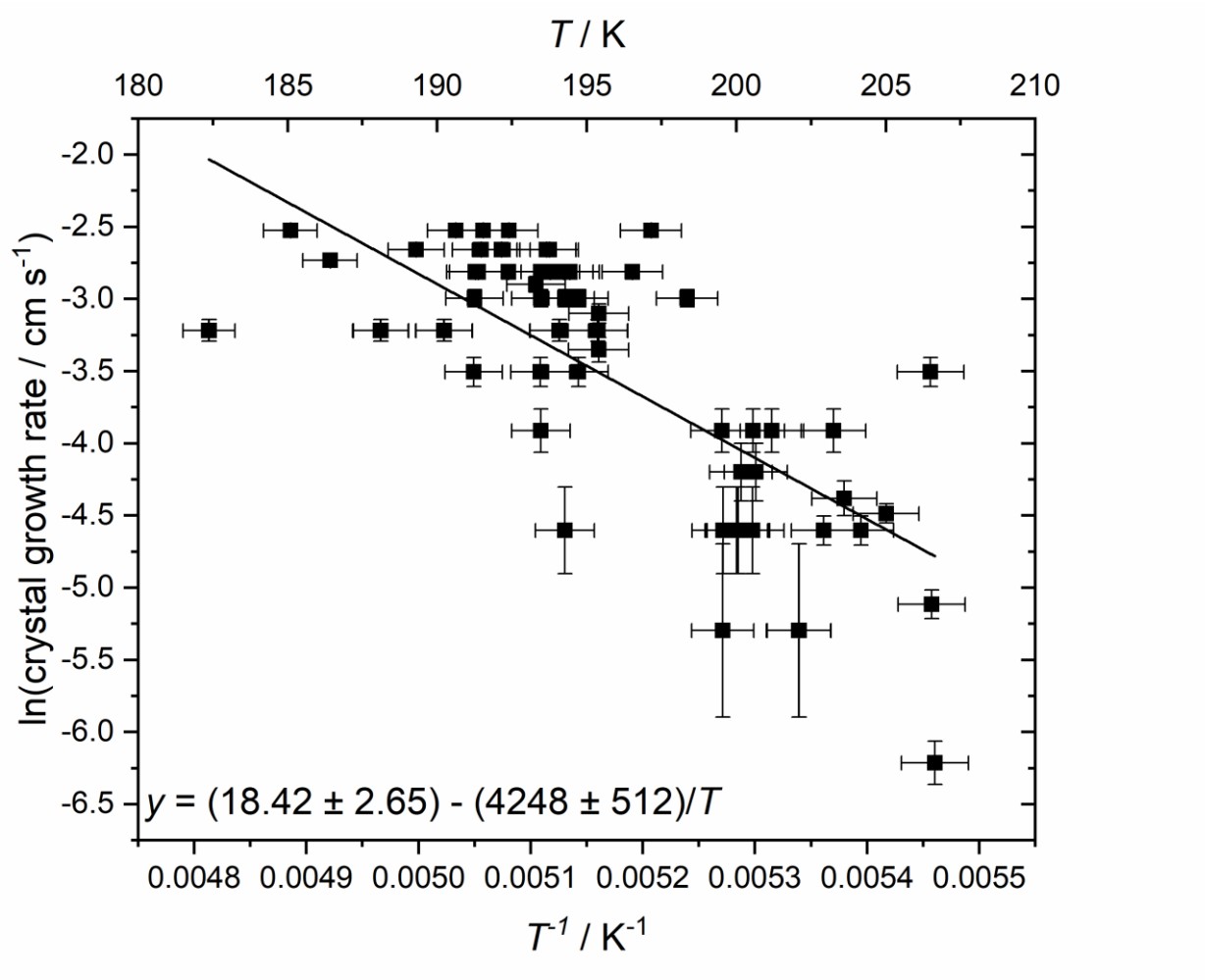

**Figure 7: Arrhenius fit to crystal growth rate to determine diffusion barrier from 40 % HNO$_3$ liquid to crystalline NAT. Horizontal error bars are a result of uncertainty in the temperature control and monitoring, vertical error bars were propagated from uncertainty in the measurement of crystal front position and the variability between video frames.**

    A number of factors may contribute to a relatively large scatter in this data. The orientation of the camera means that any

growth out of the perpendicular (to the camera) plane of the slide will be neglected, leading to an underestimate of the true

growth rate. Also, the latent heat generated will affect the droplet temperature such that there is a feedback between the droplet temperature and growth rate. However, the crystals appear to grow circularly (spherically) as opposed to e.g. cubic crystalline shapes. This implies that the diffusion of liquid material adding to the crystal is rate limiting rather than their ability to shed latent heat of crystallisation so from this perspective at least the growth rates provide a good measure of solution diffusion
energies.

## 4.2 Surface energy of a NAT cluster

The preferred method of deriving the interfacial energy between the nascent crystalline cluster and the liquid is through measurements of the temperature-dependent homogeneous nucleation of that crystalline phase (Koop and Murray, 2016). For NAX, there are relatively few measurements of homogeneous nucleation, and a number of different treatments of that data
have been used. It has been noted that extrapolation of data from homogeneous experiments to atmospheric conditions could be problematic since the dependence of the surface energy on saturation is unknown (Knopf et al., 2002). Here, rather than choosing an absolute value, we examine the values that can be derived from the literature and explore the sensitivity of our nucleation model within a range constrained by those values.

Before homogeneous nucleation measurements were available, the Turnbull correlation of the enthalpy of fusion to the surface
tension was used to derive a value for NAT (MacKenzie et al., 1998). The authors in that study derived a value of 7 kJ mol$^{-1}$ (see their Figure 2) at 200 K. To convert this to a value per surface area, we take a surface molecular density from the 110 plane of β-NAT (Wood, 1999), which has four NAT molecules in a planar unit cell 9.4845 × 14.6836 Å (Taesler et al., 1975), giving a value of 0.03 ± 0.01 J m$^{-2}$. This is a rather indirect method; and relies on speculative assumptions, such as which crystal face is growing. However, it has the advantage that since the temperature-dependent enthalpies are known, a $\sigma$ value
can be determined at atmospherically-relevant temperatures.

Homogeneous nucleation of NAT has been observed experimentally in two studies (Bertram and Sloan, 1998;Salcedo et al., 2001). In both cases the authors report measured nucleation rates but do not derive $\sigma$ values. These experimental data were used in a subsequent study to derive a $\sigma_{NAT}$ value considering the incongruent (multiple component) nature of the NAT crystal and liquid (Djikaev and Ruckenstein, 2017). Here we re-analyse the experimental rate measurements using a linearised form
of E1. We select the Tisdale et al. (1997) value for the diffusion activation energy, since it is more relevant to the concentrations used in the homogeneous nucleation experiments, the temperature and saturation data from the homogeneous nucleation experiments, and a molar volume of $5.35 \times 10^{-29}$ cm$^3$ mol$^{-1}$, based on a NAT density of 1.652 g cm$^{-3}$. The re-analysed results are shown in Figure 8, and all $\sigma_{NAT}$ values are summarised in Table 2.

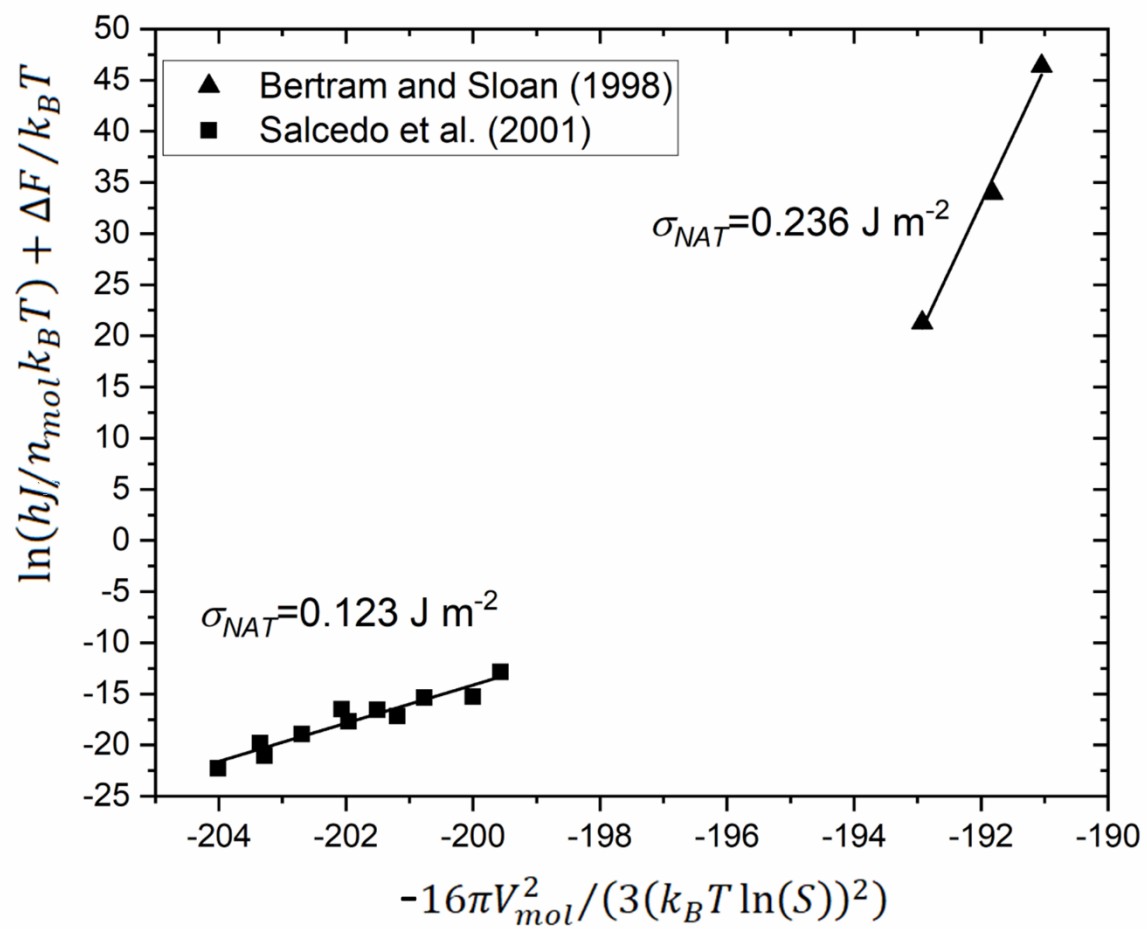

**Figure 8: Reanalysis of measured homogeneous nucleation rates from Salcedo et al. (2001) (see their Figure 7) and Bertram and Sloan (1998). Axes represent the linearised form of E4 substituted into the equivalent of E1 for homogeneous nucleation (Murray et al., 2012). Linear regression fits and derived surface energies are shown.**

These four estimates of the surface energy vary by a factor of eight. They are larger by a factor of 1.5-10 than current estimates of the value for water ice-liquid interface (Koop and Murray, 2016;Tarn et al., 2021). This seems reasonable since the interfacial energy is related to how alike the liquid and solid phases are, and NAT requires more molecules to rearrange than water ice and a greater disruption to the hydrogen bonding network at the interface. Because they were measured or derived for different conditions, and it is not known how the surface energy varies with temperature or saturation, we investigate the atmospheric implications of a range of possible interfacial energies based on these values and comparison to our experimental data on heterogeneous nucleation.

## 4.3 Contact angles

Given these physical parameters constrained by homogeneous nucleation rates, the heterogeneous activity of a substrate can then be quantified by constraining the contact angle, $\theta$, using E1-4 and measured heterogeneous nucleation rates. For MSP, our previous heterogeneous nucleation experiments using fumed silica and fused quartz are well suited to this task (James et al., 2018). We found that fused quartz (crystalline quartz which has been melted and shock frozen to give an amorphous material) with a BET surface area of 4.85 $m^2$ $g^{-1}$ and spherical equivalent particle radius of 240 nm nucleated crystallisation around 20 K warmer than a similar amount of fumed silica (a similarly amorphous material made by pyrolysis of silicon tetrachloride), which has a BET surface area of 195 $m^2$ $g^{-1}$ and spherical equivalent particle radius of 5.8 nm.

Figure 9 shows example fits of E1-4 for these two materials, using the diffusion barrier estimated from our observed temperature dependence of crystal growth rate (see Figure 7). Varying the diffusion barrier within uncertainty moves the predicted fraction frozen by several K for fused quartz, whilst the fumed silica fit is more sensitive. A range of $\sigma_{NAT}$ values constrained by the homogeneous measurements were considered. Because this value is not well established, multiple solutions for $\theta$ are possible. For each $\sigma_{NAT}$ value the experimental conditions of the fused quartz data were first used to find a $\theta$ which gave good agreement with the observed nucleation temperatures. The radius of the nucleating particle, $r_{NP}$, was taken as the spherical equivalent of the measured BET surface area; however, for fused quartz the calculated nucleation rate is insensitive to this quantity as the nucleating particle is significantly larger than the critical cluster size. This gave a paired $\theta$ for each $\sigma_{NAT}$, as shown in Table 2, and for $\sigma_{NAT}=0.1$ J $m^{-2}$ as an example in Figure 9. The calculated nucleation rate is rather sensitive to $\theta$, so the heterogeneous activity of the amorphous silica material could be quantified within a small range. We assigned a single contact angle to describe nucleation on all areas of each of these silica surfaces. The fact that the CNT curves in Fig 9 reproduce the steepness of the experimental curves implies that this is a reasonable assumption, despite the fact that in many heterogeneous nucleating systems a distribution of contact angles is required to describe the data (Herbert et al., 2014). Making the assumption that the fumed silica has similar nucleation properties to fused quartz, and that its different nucleation temperatures are due only to its smaller particle size, we then use these paired $\theta$ and $\sigma_{NAT}$ values with a varied $r_{NP}$ to reproduce the observed nucleation rates of fumed silica suspensions. The resulting particle sizes, shown in Table 2, give a further check on the physical reasonability of the thermodynamic data. Some variability from the size estimated from BET is reasonable: firstly because this estimate assumes that particles are uniformly spherical; and secondly because the silica can partially dissolve and reprecipitate in acid solution to give a change in particle size.

Figure 9 shows self-consistency between measurements of nucleation activity on these two chemically similar materials. This suggests that size-dependent CNT is a good model for this process, and that the laboratory measurements of heterogeneous nucleating activity are a good way to constrain the physical system and thereby quantify the atmospheric process.

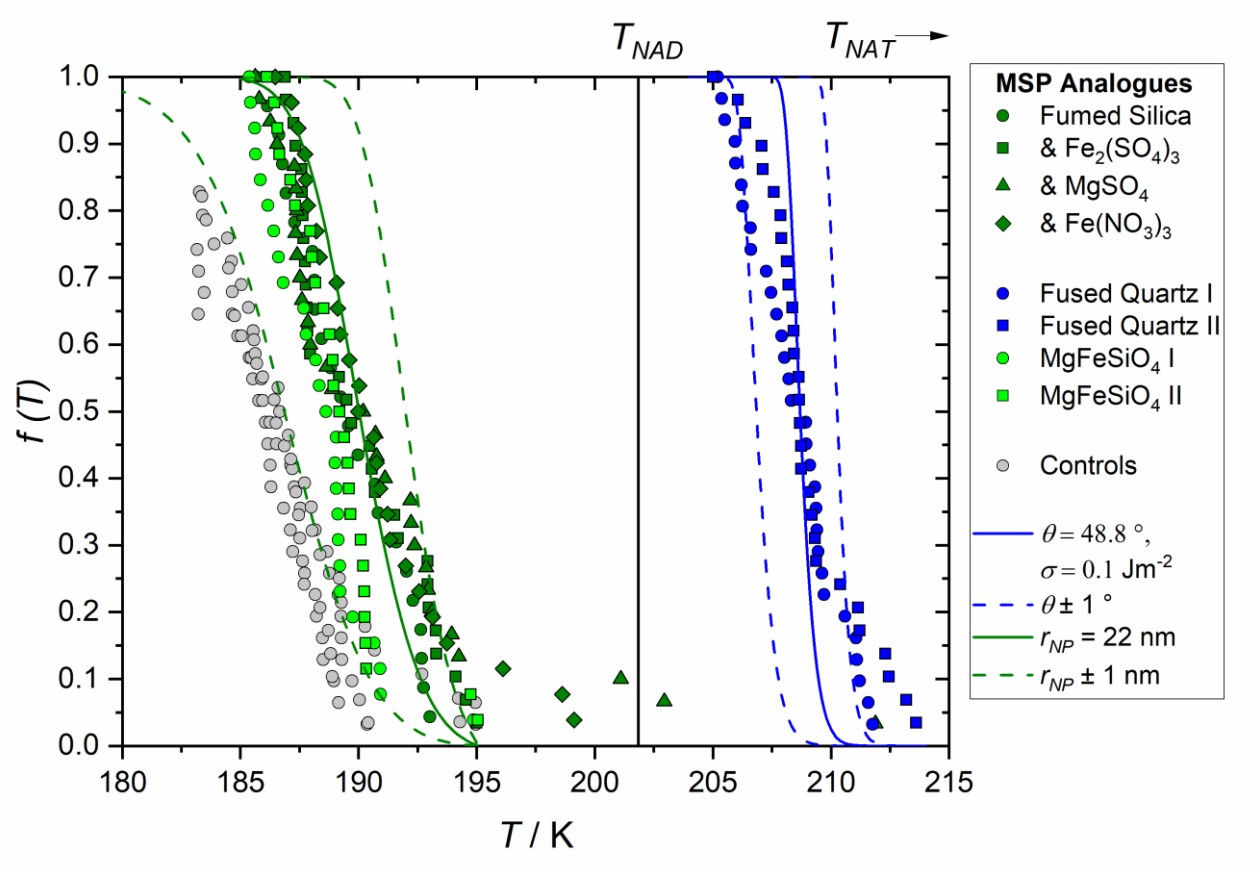

**Figure 9: Meteoric Smoke Particle (MSP) analogue fraction frozen as measured in our previous work (points, James et al. (2018)) and as modelled here using the size dependent nucleation rate parameterisation described by equations E1 to E4, lines. Grey points show the instrument background. Blue data shows nucleation measured on fused quartz (BET surface area 4.85 $m^2$ $g^{-1}$) and green data fumed silica (BET surface area 195 $m^2$ $g^{-1}$). Fused quartz data is used to constrain the contact angle for a given surface energy since this is insensitive to particle size, blue solid line shows 0.1 J $m^{-2}$ as an example, and blue dashed lines show the sensitivity to the contact angle. These paired $\theta$ and $\sigma$ values are then used to determine the nucleating particle radius, $r_{NP}$, which gives a good fit (solid green line) to the fumed silica data, again shown here for $\sigma_{NAT}$=0.1 J $m^{-2}$, dashed green lines show the sensitivity to the particle size.**


**Table 2: Surface energy, $\sigma_{NAT}$, values in the literature or derived from literature nucleation rate data**

| $\sigma_{NAT}$ / J m$^{-2}$ | Valid $T$ / K | Valid [HNO₃] /wt% | $\theta$ from fused quartz experiments/ ° | $r_{NP}$ from fumed silica / nm | reference |
|---|---|---|---|---|---|
| 0.03 | 200 | 54 | unphysical | N/A | MacKenzie et al. (1998) |
| 0.11 | 155-180 | 50-64 | 45 | 23 | Djikaev and Ruckenstein (2017) |
| 0.123 | 175-180 | 50-64 | 41 | 24 | Derived here from Salcedo et al. (2001) |
| 0.236 | 155-175 | 54 | 24.4 | 27 | Derived here from Bertram and Sloan (1998) |
| Reasonable range of values chosen here for atmospheric comparison: | | | | | |
| 0.05 | 210-185 | 40 | 100 | 4 | |
| 0.1 | 210-185 | 40 | 48.8 | 22 | |
| 0.15 | 210-185 | 40 | 34.9 | 25 | |

We now explore the sensitivity of the parameterisation to the input parameters. The interfacial energy value determined by the Turnbull correlation resulted in a nucleation rate too fast at temperatures warmer than 215 K to explain the observed fused quartz nucleation data, even assuming homogeneous nucleation. We therefore recommend a lower limit value for $\sigma_{NAT}$ of 0.05 J m$^{-2}$, which produces a good fit to both amorphous silica datasets with $\theta = 100$ ° and $r_{NP} = 4$ nm, 30% smaller than the BET spherical equivalent size of the fumed silica. Both our and Djikaev and Ruckenstein (2017)'s analysis of homogeneous nucleation measured by Salcedo et al. (2001) place $\sigma_{NAT}$ between 0.1 J m$^{-2}$ and 0.15 J m$^{-2}$. The value we derive from the measurements of Bertram and Sloan (1998) is 0.236 J m$^{-2}$, somewhat higher than the other constraints. We do not test a range of values which includes our analysis of the Bertram and Sloan (1998) data since there are fewer data points on which to base this value and controlling the conditions in flow tube experiments such as those of Bertram and Sloan (1998) is known to be challenging. Such a large $\sigma_{NAT}$ would require a very active (small contact angle) amorphous silica material to explain our observed nucleation rates. For example, using a slightly different application of CNT, Hoyle et al. (2013) found that a minimum contact angle of 43° gave a good agreement with observed cloud. Note that we do not rule out the possibility that $\sigma_{NAT}$ might be this high, but we consider it unlikely. We therefore go on to examine the atmospheric implications of this model of the

heterogeneous NAT nucleating activity of amorphous silica from MSPs, investigating a range of interfacial energy values from
0.05 to 0.15 $Jm^{-2}$, and finally compare the likely atmospheric impacts of MSPs and fragments.

## 4.4 The likely nature of MSP in ternary acid solutions

It is well established that acid processing of MSPs results in dissolution of most metal components, leaving silica and alumina solids in suspension (Murphy et al., 2014;Saunders et al., 2012). Indeed the similar activity of synthetic MSP analogues to fumed silica measured in James et al. (2018) suggests that acid processing leaves these materials alike.

In the presence of $H_2SO_4$, these suspensions were found to form a gel within several hours at room temperature, which showed no nucleation activity above the instrument baseline (data not shown). Silica suspensions here contained approximately 2.5 wt% silica, around a factor of five larger than atmospheric concentrations (Cziczo et al., 2001). Silica gelation is also known to be strongly temperature dependent (Colby et al., 1986). This suggests that this is an artifact of the laboratory method, and that stratospheric droplets would not form a gel. Indeed, if the silica particles formed a gel throughout the droplet, the silica
would be evenly distributed and atmospheric single particle mass spectrometers would detect a narrow distribution of ion ratios for silicon, as they do for e.g. iron and nickel (Murphy et al., 2014). Electron microscopy of collected particles has shown the presence of agglomerated spherical particles, which is also not consistent with gel formation (Ebert et al., 2016;Bigg, 2012).

## 4.5 Atmospherically available MSP

To implement this size dependent parameterisation of heterogeneous NAT nucleation by amorphous silica from MSPs, we
combine modelling of atmospheric MSP chemistry and transport with observed aerosol size distributions. The most recent estimate of the ablated meteoroid mass, which provides the material from which MSPs form, is 8.3 tons $day^{-1}$ (Carrillo-Sánchez et al., 2020). Modelling of the growth, atmospheric circulation and entrainment of these particles in stratospheric sulfate aerosol suggests an average mass concentration of $(1.5 \pm 0.5) \times 10^{-15}$ g $cm^{-3}$ at 67 °N latitude, 70 hPa altitude in February (James et al., 2018). We then consider a size distribution as measured for particles collected in the lower and middle stratosphere, which
were found to have little variation with altitude (Bigg, 2012). By using the observed size distribution to define the smoke particle size distribution and normalising to the mass concentration from WACCM modelling, we obtain the "initial" size distribution shown in Figure 10. This distribution contains an integrated number concentration of 23.5 particles $cm^{-3}$, similar to commonly applied estimates of around 20 $cm^{-3}$ for the number of liquid aerosol in the stratosphere (Hoyle et al., 2013). The integrated particle surface area is 0.08 $\mu m^2$ $cm^{-3}$, approximately a factor of two less than applied by assuming mono-disperse
particles in James et al. (2018).

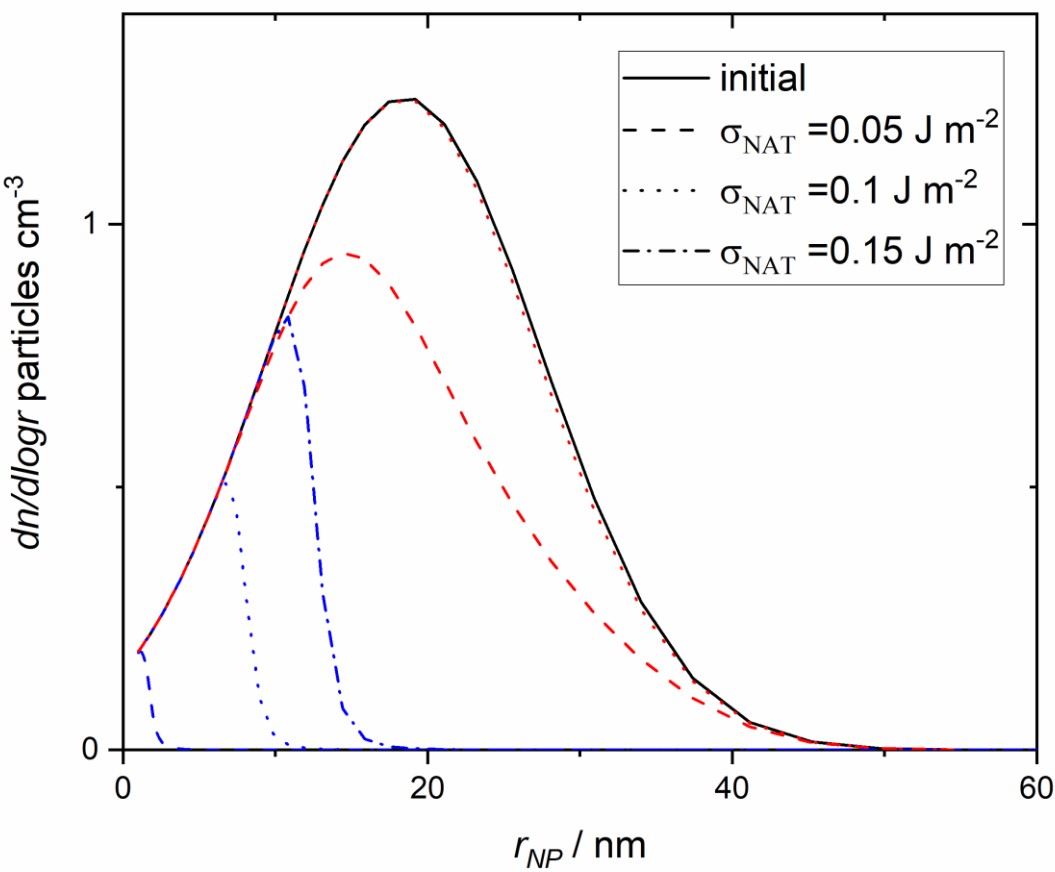

**Figure 10: Size distribution of MSPs. Initial (black, solid line) distribution and that of the particles that "survive" (do not cause nucleation) is shown. Line style differentiates assumed surface energy values as indicated in the legend, blue lines show values assuming 15 ppbv HNO₃ and red lines show values with 10 ppmv. The surviving distribution assuming 10 ppbv HNO₃ and $\sigma_{NAT}$ = 0.15 is indistinguishable from the initial distribution on this scale.**

### 4.6 Simulated atmospheric nucleation by MSPs

We now combine this size-dependent nucleation parameterisation with the distribution of available MSPs and the same atmospheric trajectory model we applied previously (James et al., 2018). The nucleation rates and resulting crystal number concentrations are calculated by assuming that each MSP occupies a separate liquid droplet, *i.e.* that each can cause a single nucleation event. We do not account for processes such as the transfer of material from surviving droplets to the nucleated crystals, or the sedimentation of the growing crystals from the modelled volume. As long as the equilibrium vapour pressure over the NAT crystal is lower than over the droplet (*i.e.* the atmosphere is saturated with respect to NAT), there will be a net

transfer of HNO$_3$ from the remaining liquid to the growing crystal. This results in reduced HNO$_3$ concentration and saturation in the liquid, reducing subsequent nucleation. The resulting crystals can grow to micron or even 10s micron scales, causing

them to sediment out of the NAT saturated airmass (Fueglistaler et al., 2002). These cloud micro-physical processes require a detailed microphysical model to address, hence are out of scope here. Carslaw et al. (2002), found that a time step of 30 minutes or less was required to accurately account for these processes in clouds with particle concentrations of $10^{-4}$ cm$^{-3}$ or less, so their effects could be significant across the 18 hours of our trajectory model. By neglecting this we can only produce an upper limit to the number of crystals which could form in the atmosphere. As a result of this limitation, we aim here only to decide

whether MSPs could be active enough to explain the observations, not to exactly reproduce them.

The results are compared to observed crystal concentrations in the atmosphere (Voigt et al., 2005), and other parameterisations of nucleation, in Figure 11. Assuming 15 ppmv HNO$_3$, more than half of the MSP particles are able to nucleate NAT within the trajectory timescale. The surviving distribution of MSPs (which do not cause nucleation within this trajectory model) is shown by dashed lines in Figure 10, and demonstrates that when the size of the nucleating particle is taken into account, there

is essentially a threshold below which particles are too small to cause nucleation. The threshold size depends on the value taken for the interfacial energy. For 10 ppbv HNO$_3$, less nucleation is observed, though crystal number concentrations still significantly exceed those measured in the atmosphere. The final number concentration of crystals is a strong function of the chosen value of $\sigma_{NAT}$.

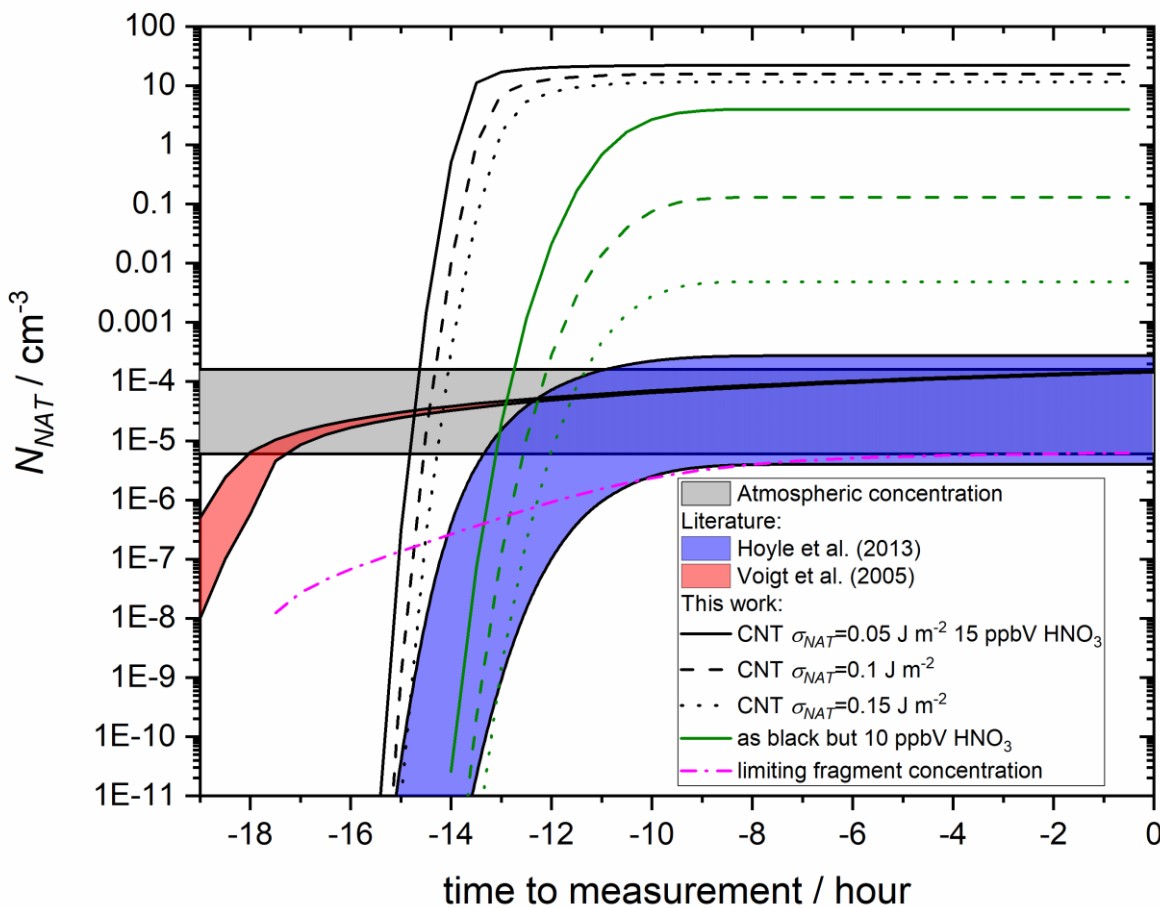

**Figure 11: NAT particle production using a temperature profile based on stratospheric observations from Voigt et al. (2005). See James et al. (2018) Figure 6(a) for corresponding temperature and saturations. $S_{NAT}$ were calculated at 70 hPa assuming 5 ppmV $H_2O$, 0.1 ppbV $H_2SO_4$ and 10 (minimum of ranges and blue lines) to 15 (maximum of ranges and black lines) ppbV $HNO_3$. Growth, sedimentation of particles and removal of $HNO_3$ are not taken into account. These processes will limit the number of NAT particles that can nucleate by reducing the $HNO_3$ concentration and NAT saturation in remaining droplets, hence our predicted NAT number**
**concentrations are an upper limit. Predicted $N_{NAT}$ based on nucleation parameterisations from this work compared to James et al. (2018) and estimated surface areas and size distributions of MSPs as well as two literature parameterisations (Voigt et al., 2005;Hoyle et al., 2013). The CNT parameterisation produced in this work is shown for a range of surface energy ($\sigma_{NAT}$) values differentiated by line type, using paired contact angles constrained by heterogenous nucleation experiments. Crystal number concentrations calculated from a parameterisation of meteoric fragments are also shown, with the availability of fragments varied to reproduce the**
**lowest observed atmospheric concentration.**

The nature of MSPs in atmospheric liquid droplets may also contribute to this overestimation. Here we assume that the particles

in our size distribution are dispersed evenly and each particle is able to nucleate a crystal. In fact this distribution relates to the

size of primary grains that were observed as agglomerates in atmospheric droplets (Bigg, 2012). This means that more than

one nucleating particle would be removed with each nucleation event, and the final number of crystals would be lower. Voigt et al. (2005) measured a concentration of 10 particles cm$^{-3}$ (liquid droplets) on the flights to which these back trajectories relate. A number of studies have shown that typically 50-80 % of stratospheric liquid aerosol contains a refractory core (Weigel et al., 2014;Schneider et al., 2021;Murphy et al., 2021). This would suggest that at least 5 particles cm$^{-3}$ contained an MSP, with an average of 4.7 primary particles per droplet, randomly sampled from the total distribution. This would mean that any nucleation event would remove all primary particles in that liquid droplet, with a linear reduction in the final number concentration of crystals. For the trajectory calculated using 10 ppbV HNO$_3$ and $\sigma_{NAT}$ = 0.15, this would result in 10$^{-3}$ crystals cm$^{-3}$, still significantly higher than the measured concentration.

We conclude that nano-scale amorphous silica particles formed by dissolution of MSPs in stratospheric sulfuric acid droplets are sufficiently active heterogeneous nucleators of NAT to explain observed cloud crystallisation warmer than the H$_2$O ice melting point. This could have significant implications for particularly early season Antarctic or Arctic clouds where temperatures may not be cold enough for water ice to form, and hence impact on the buildup of ozone-destroying species throughout the winter and eventually on ozone depletion.

We recommend that the CNT parameterisation presented here is deployed in atmospheric models of PSC crystallisation and ozone depletion and the effect of varying the NAT/liquid interfacial energy investigated. In a model that described droplet-to-crystal material transfer, particle growth and sedimentation, the number of crystals formed would reduce towards the observed values. Additionally, if our observation that the number of crystals formed is influenced by the input value of $\sigma_{NAT}$ holds, the "atmospheric laboratory" may now be an effective domain to further constrain this important thermodynamic quantity. The system is complex, with remaining uncertainties in a number of thermodynamic quantities, so that it is difficult to predict which effect will control the number of crystals formed under any given atmospheric conditions. A key remaining task is to test the effect of H$_2$SO$_4$ on the nucleation ability of MSP analogues. We were not able to test this due to the observation that ternary suspensions of silica form gels rapidly under laboratory conditions. Other experimental approaches such as cloud chamber experiments may be better suited to investigate this sensitivity (Wagner et al., 2005b).

## 5. Conclusions

In the laboratory, analogues for meteoric fragments can nucleate nitric acid hydrate (NAX) crystals in ternary solution droplets with compositions relevant to Polar Stratospheric Clouds (PSCs). This nucleation shows complex behaviour, but in some cases is resistant to deactivation by both nitric and sulfuric acids in atmospheric concentrations. This was shown previously (James et al., 2018), but we have now extended these studies to include sulfuric acid and annealing of meteoric fragments at temperatures experienced during entry into the atmosphere. However, consideration of the latest understanding of meteoric fragmentation and sedimentation processes suggest that there is unlikely to be sufficient input flux of fragmenting meteoroids in order for this material to heterogeneously nucleate observed crystal numbers in PSC. Hence, while any meteoric fragments

could nucleate Nitric Acid Trihydrate (NAT) in the polar stratosphere there are unlikely to be sufficient numbers to nucleate the majority of observed NAT crystals under any conditions and we conclude some other nucleation pathway must dominate.

We propose that acid processed Meteoric Smoke Particles (MSPs, i.e. silica) can nucleate NAT, despite its small size. We constrain a model of nucleating particle size-dependent classical nucleation theory by combining existing laboratory data on diffusion barriers under homogeneous conditions, surface free energy, heterogeneous nucleation activity of amorphous silica materials, and a new measurement of diffusion barriers under heterogeneous nucleation conditions. Application of this parameterisation to atmospheric cloud observations overpredicts the resulting crystal number densities. The comparison carried out here uses state-of-the art knowledge of MSPs in the atmosphere, which appears robust, but neglects the growth and sedimentation of particles after nucleation, which would reduce the total crystal number. This suggests that MSPs are sufficiently active to explain observed crystal numbers in polar stratospheric clouds, and that application of our constrained nucleation activity in more complete atmospheric models could now provide an improved understanding of PSC microphysics and ultimately ozone depletion.

This work shows that the modelling of crystal nucleation in early season PSCs and the resulting ozone depletion relies on the nucleation of NAX crystals on MSPs. Hence, in order to quantitatively predict the effect of a changing climate, long-term ozone recovery or events such as volcanic eruptions on stratospheric ozone an adequate understanding of the meteoric input, the sources of meteoric material in the solar system and how these interact with Earth's atmosphere, the production of MSPs in the mesosphere and its transport through the mesosphere and stratosphere are all needed.

**Acknowledgements**

This work forms part of the MeteorStrat project funded by the UK Natural Environment Research Council (grant number NE/R011222/1).

**Code / data availability**

Figure data is available from the University of Leeds research repository [DOI to be created on acceptance].

**Author Contribution**

ADJ performed laboratory experiments, carried out data analysis and led paper drafting.

FP and SNFS assisted with laboratory experiments, instrument design and data analysis.

GWM advised on atmospheric modelling and is project PI.

BJM and JMCP supervised laboratory work, analysis and participated in paper drafting.

## Competing interests

The authors declare no competing interests.

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
