# Peer review of "The importance of acid processed meteoric smoke relative to meteoric fragments for crystal nucleation in polar stratospheric clouds"

_Atmospheric Chemistry and Physics, 2022_

## Referee Comment (RC1)

**Referee Report on manuscript acp-2022-598 "The importance of acid processed meteoric smoke relative to meteoric fragments for crystal nucleation in polar stratospheric clouds" by A.D. James, F. Pace, S.N.F. Sikora, G.W. Mann, J.M.C. Plane and B.J. Murray submitted to Atmospheric Chemistry and Physics Discussions**

The present work deals on the one hand with the old, but as yet unanswered question about the nucleation rate of PSC I (NAT) clouds that are necessary, together with ice particles (PSC II) and other $H_2SO_4$-containing aerosols (frozen ternary solutions or glasses), to explain the rapid and recurrent seasonal stratospheric ozone loss, both in the Antarctic and, to a lesser extent, also in the Arctic. In this paper nucleation rates are measured in deposited microdroplet plate experiments under satisfactory temperature control using visual observation on the fraction of frozen droplets as a function of temperature. On the other hand, the paper deals with the application of Classical Nucleation Theory (CNT) to literature data (James, 2018) and to results obtained in this study starting with section 4 of the present paper. CNT is a formally thermodynamic network cast into a kinetic theory (akin to transition state theory) that uses scarce, scattered and inconsistent macroscopic data such as contact angles and surface energies that do not constrain anything, but leaves the bewildered reader on a heap of unexplained and less than intuitive physical pictures. Even though I concur with most if not all of the conclusions of the authors that are deemed to be plausible there seems to be a disconnect between the two parts which seems to be a sign of incomplete "integration" of the two halves of this publication. The paper is well written, but sometimes needs additional explanations in order for the reader to grasp the experimental development and evaluation of the data (see below). I believe that the paper, if reworked and partially reformulated, would be of significant interest to the community of atmospheric chemists and numerical modelers trying to understand the physical-chemical basis and circumstances such as the locus of nucleation of PSC I clouds because several interesting facts about nucleation are presented, both in the first experimental and further in the second theoretical (CNT) section. The second modeling part needs to be streamlined, but at the same time has to be linked to intuitive and plausible concepts (see below). I am willing to support the publication of this two-pronged paper once the authors have had a chance to address my several questions and suggestions in a revised manuscript. However, I request to inspect the improvements and alterations before this publication is given the green light to proceed.

Here are my main points I would like to raise in the order of the presentation of the text:

- Addressing the first part of the manuscript a discussion of experimental uncertainties is nowhere to be found throughout the manuscript. Every scientific quantification needs one in view of the many parameters that control the measured nucleation rate. From Figure 2 it is apparent that per run 40 droplets in total are considered leading to a minimum of the random error of observation at the 50% point of fraction frozen. The experimental error at the fringes of the measurement domain become larger, and it behooves the authors to label a random error to selected data points such as displayed in Figure 3.
- Regarding the method I wonder about the gas-tightness of the cover plate swept with dry nitrogen gas during measurements. Have the authors performed tests in this regard? If gas-tightness would be only partial it may be conceivable that the $HNO_3$ and/or $HNO_3/H_2SO_4$ concentration would increase over time because the partial pressure of $H_2O$ is much larger than that of $HNO_3$, and even more so for $H_2SO_4$ according to the pertinent phase diagram. Have any attempts at measuring the pH value before and at the end of each experimental run been undertaken? Or before and after adding the meteoric fragments (line 148)! Owing to the high acidities the authors probably would have to work with Hammett acidity functions (possibly negative pH values).

- The symbols, especially referring to the control runs in Figure 3, are too faint and all characters are too small (adjust font size) to read-off the given chart. Please anticipate the drawing in print! Line 182 mentions a factor of ten in nucleation temperature when adding $H_2SO_4$. I for one am able to "find" at most a factor of five difference, so I request that the authors specify which points they are comparing in Figure 3. Regarding Figure 3 I observe a certain irreversible change in nucleation behavior in consecutive runs. The authors never venture out into a possible reason for this behavior when going from run 1 to 2 to 3.

- On line 137 the authors mention an average of 18 $\mu m$ (microns) dimension of the added meteoric material. Do they have any idea about size dispersion and shapes from imaging of meteoric material? Any BET measurement for total external and internal surface of the used material? How do the authors proceed to evaluate $n_s$, the number of active sites per unit surface area of solid inclusion? This screams for an explanation and in my view is a show-stopper when presenting Figure 4. What is the definition of $n_s$ used, and what is it based on?

- One of the most interesting aspects of this work is the conversion of the relative results presented in Figure 3 to absolute values in Figure 4. The authors have to stepwise explain how they arrive at the surface area of 0.2 $\mu m^2$ $cm^{-3}$ as a minimum surface area from their data in relation to the observed crystal concentration of 6 $10^{-6}$ $cm^{-3}$. The authors do not explain where these numbers emerge from. Please oblige as the connection to absolute numbers is an important aspect of the results as the primary observable (fraction frozen) is a relative finding.

- A semantic point: Lines 288 and 293 mention "meteoroids"? What is the definition compared to meteoric materials? Why do you need to use an additional terminology here?

Regarding Section 4 onwards I have only two questions:

- The authors repeatedly write about size-dependent effects on parameters feeding into macroscopic CNT such as $\sigma_{NAT}$, $\theta$, $r_{NP}$. As far as I see it the only size-dependent parameter is $\sigma_{NAT}$ from which all size-dependence follows according to equations (1) to (4) in the Introduction. Is this equivalent to the effect of curvature of ever smaller particles that becomes dominant for sizes below 20 nm or so when evaluating the surface energy $\sigma_{NAT}$? From what the authors tell me I am not sure about it as they never attribute the size-dependence to any physical or geometrical parameter. If this is the case the next step would be to point out the conundrum of CNT for small (curved) particles as the classical picture of a smooth envelope fails owing to the molecular view of a surface with its molecular asperities. In any case, Table 2 yields instructive examples which the authors should comment on in depth in order to enhance understanding of the CNT results. Their treatment reminds me of Fletcher theory of touching spheres of ice in order to describe capillarity effects and condensation phenomena.

- One of the main conclusions of this work is that quartz particles of several hundred nm's and possibly silicates do not lend themselves as nucleating substances for PSC I formation over the poles owing to insufficient numbers whereas much smaller and more numerous MSP particles would be the favored nucleating substrate. In view of the fact that by far the most abundant atmospheric dust source is from terrestric crustal material whipped up by winds could it be that the fine fraction of this atmospheric dust could possible be responsible for PSC I nucleation in the stratosphere after some strat-trop exchange? In contrast the Meteoric Input Function (MIF) is highly uncertain as the authors convincingly point out. A statement by the authors admitting or refuting this terrestrial source as well as the reasons for it would be helpful to clarify the situation.

- The Herbert et al., 2015 reference is missing. The last reference Wood, S.E. is incomplete.

---

## Author Response (AR1)

The authors are grateful for these thorough and highly relevant reviewer comments. We have responded to these below and made revisions throughout the manuscript. Sections of the manuscript that have been significantly changed are identified with blue font, though other minor changes such as abbreviations have been made but are not highlighted.

Reviewer 1

The present work deals on the one hand with the old, but as yet unanswered question about the nucleation rate of PSC I (NAT) clouds that are necessary, together with ice particles (PSC II) and other $H_2SO_4$-containing aerosols (frozen ternary solutions or glasses), to explain the rapid and recurrent seasonal stratospheric ozone loss, both in the Antarctic and, to a lesser extent, also in the Arctic. In this paper nucleation rates are measured in deposited microdroplet plate experiments under satisfactory temperature control using visual observation on the fraction of frozen droplets as a function of temperature. On the other hand, the paper deals with the application of Classical Nucleation Theory (CNT) to literature data (James, 2018) and to results obtained in this study starting with section 4 of the present paper. CNT is a formally thermodynamic network cast into a kinetic theory (akin to transition state theory) that uses scarce, scattered and inconsistent macroscopic data such as contact angles and surface energies that do not constrain anything, but leaves the bewildered reader on a heap of unexplained and less than intuitive physical pictures. Even though I concur with most if not all of the conclusions of the authors that are deemed to be plausible there seems to be a disconnect between the two parts which seems to be a sign of incomplete "integration" of the two halves of this publication. The paper is well written, but sometimes needs additional explanations in order for the reader to grasp the experimental development and evaluation of the data (see below). I believe that the paper, if reworked and partially reformulated, would be of significant interest to the community of atmospheric chemists and numerical modelers trying to understand the physical-chemical basis and circumstances such as the locus of nucleation of PSC I clouds because several interesting facts about nucleation are presented, both in the first experimental and further in the second theoretical (CNT) section. The second modeling part needs to be streamlined, but at the same time has to be linked to intuitive and plausible concepts (see below). I am willing to support the publication of this two-pronged paper once the authors have had a chance to address my several questions and suggestions in a revised manuscript. However, I request to inspect the improvements and alterations before this publication is given the green light to proceed.

Here are my main points I would like to raise in the order of the presentation of the text:

1.1 Addressing the first part of the manuscript a discussion of experimental uncertainties is nowhere to be found throughout the manuscript. Every scientific quantification needs one in view of the many parameters that control the measured nucleation rate. From Figure 2 it is apparent that per run 40 droplets in total are considered leading to a minimum of the random error of observation at the 50% point of fraction frozen. The experimental error at the fringes of the measurement domain become larger, and it behooves the authors to label a random error to selected data points such as displayed in Figure 3.
The authors agree that error analysis is important. We had presented multiple runs conducted under the same conditions to indicate uncertainty, but had failed to discuss this properly. In addition to adding a brief discussion of uncertainties (ln 265) we have also added error bars to represent several important source of uncertainty in Figure 4 for example datasets. We have also added error bars to Figure 7 and amended the caption to Figure 3 to include the fixed temperature uncertainty.

1.2 Regarding the method I wonder about the gas-tightness of the cover plate swept with dry nitrogen gas during measurements. Have the authors performed tests in this regard? If gas-tightness would be only partial it may be conceivable that the $HNO_3$ and/or $HNO_3/H_2SO_4$ concentration would increase over time because the partial pressure of $H_2O$ is much larger than that of $HNO_3$, and even more so for $H_2SO_4$ according to the pertinent phase diagram. Have any attempts at measuring the pH value before and at the end of each experimental run been undertaken? Or before and after adding the meteoric fragments (line 148)! Owing to the high acidities the authors probably would have to work with Hammett acidity functions (possibly negative pH values).

Measuring the melting point provides a test of both the temperature measurement and the concentration of the droplet. If the variation in melting point were entirely due to changes in $HNO_3$ concentration, the variation from the initial concentration would be $0.5 \pm 0.9$ wt %. Note that some measured melting points are below that expected for the initial concentration where, as the reviewer correctly identifies, a poor seal would tend to increase the concentration and therefore the melting point. As a result we believe that the majority of the variation in melting point is due to uncertainty in the temperature control and measurement, which is also reasonable compared to uncertainties reported with similar experimental apparatus (Whale et al., 2015). This discussion has been expanded in the revised draft (line 205).

1.3 The symbols, especially referring to the control runs in Figure 3, are too faint and all characters are too small (adjust font size) to read-off the given chart. Please anticipate the drawing in print! Line 182 mentions a factor of ten in nucleation temperature when adding $H_2SO_4$. I for one am able to "find" at most a factor of five difference, so I request that the authors specify which points they are comparing in Figure 3. Regarding Figure 3 I observe a certain irreversible change in nucleation behavior in consecutive runs. The authors never venture out into a possible reason for this behavior when going from run 1 to 2 to 3.

Line thickness and colour and font sizes on Figures 3 and 4 have been adjusted to improve readability. The previous text was imprecise, and has been revised (line 225) to stress the key point that in most cases the Allende samples with $H_2SO_4$ do not show nucleation activity above the instrument baseline. We have also added discussion of why the sample nucleation activity might change on acid processing, however we note that this observation is not critical to the aims or conclusions of this work.

1.4 On line 137 the authors mention an average of 18 mm (microns) dimension of the added meteoric material. Do they have any idea about size dispersion and shapes from imaging of meteoric material? Any BET measurement for total external and internal surface of the used material? How do the authors proceed to evaluate $n_s$, the number of active sites per unit surface area of solid inclusion? This screams for an explanation and in my view is a show-stopper when presenting Figure 4. What is the definition of $n_s$ used, and what is it based on?

Omission of this discussion was an error which we are happy to have corrected. BET surface areas have been added to Table 1, and this was clarified both in the method (line 171) and results (line 258) sections.

1.5 One of the most interesting aspects of this work is the conversion of the relative results presented in Figure 3 to absolute values in Figure 4. The authors have to stepwise explain how they arrive at the surface area of 0.2 mm$^2$ cm$^{-3}$ as a minimum surface area from their data in relation to the observed crystal concentration of 6 $10^{-6}$ cm$^{-3}$. The authors do not explain where these numbers emerge from. Please oblige as the connection to absolute

numbers is an important aspect of the results as the primary observable (fraction frozen) is a relative finding.

The discussion of this method has been expanded (line 290).

1.6 A semantic point: Lines 288 and 293 mention "meteoroids"? What is the definition compared to meteoric materials? Why do you need to use an additional terminology here?

Whilst technically correct, this was an inconsistency in our terminology, which we have revised. In the introduction we have defined the term "meteoroid". We now consistently use the term "meteoric fragment" for this particular stratospheric aerosol population throughout the text. Meteoric material is an umbrella term for both meteoric fragments and meteoric smoke particles.

Regarding Section 4 onwards I have only two questions:

1.7 The authors repeatedly write about size-dependent effects on parameters feeding into macroscopic CNT such as $sigma_{NAT}$, theta, $r_{NP}$. As far as I see it the only size-dependent parameter is $s_{NAT}$ from which all size-dependence follows according to equations (1) to (4) in the Introduction. Is this equivalent to the effect of curvature of ever smaller particles that becomes dominant for sizes below 20 nm or so when evaluating the surface energy $sigma_{NAT}$? From what the authors tell me I am not sure about it as they never attribute the size-dependence to any physical or geometrical parameter. If this is the case the next step would be to point out the conundrum of CNT for small (curved) particles as the classical picture of a smooth envelope fails owing to the molecular view of a surface with its molecular asperities. In any case, Table 2 yields instructive examples which the authors should comment on in depth in order to enhance understanding of the CNT results. Their treatment reminds me of Fletcher theory of touching spheres of ice in order to describe capillarity effects and condensation phenomena.

We are indeed proposing that the work of Fletcher is relevant here, and have added direct reference to that work as well as expanded discussion (line 366). Discussion of the values in Table 2 has been moved after the table for clarity.

1.8 One of the main conclusions of this work is that quartz particles of several hundred nm's and possibly silicates do not lend themselves as nucleating substances for PSC I formation over the poles owing to insufficient numbers whereas much smaller and more numerous MSP particles would be the favored nucleating substrate. In view of the fact that by far the most abundant atmospheric dust source is from terrestric crustal material whipped up by winds could it be that the fine fraction of this atmospheric dust could possible be responsible for PSC I nucleation in the stratosphere after some strat-trop exchange? In contrast the Meteoric Input Function (MIF) is highly uncertain as the authors convincingly point out. A statement by the authors admitting or refuting this terrestrial source as well as the reasons for it would be helpful to clarify the situation.

Firstly, both quartz and silica samples used here were intended as representative analogues of MSP. In contrast, meteoric fragments were represented by ground meteorites since they likely have a complex and heterogeneous composition similar to the incoming material. This has been clarified in the introduction (line 70) and methods (line 165 and 189) sections.

Terrestrial aerosol is indeed present in much of the stratosphere; however, during PSC season there is a much enhanced concentration of meteoric material due to the downwelling air from the mesosphere. This was discussed at length in the recent Kremser et al. review. We have expanded this discussion in the introduction (line 50).

The Herbert et al., 2015 reference is missing. The last reference Woods, S.E. is incomplete.
Line 649: "metals"

These errors have been corrected.

Reviewer 2

The paper by Alexander James and colleagues addresses an important and highly up to date topic. The presence, the amount, and the importance of meteoric material in the stratosphere is uncertain but discussed in recent studies to explain observations of polar stratospheric clouds (PSCs). Any piece of information from measurements and laboratory studies is welcome to solve this puzzle. The study by James et al. delivers such new and interesting findings supporting ideas about heterogeneous cloud formation on meteoric material in the polar winter stratosphere. However, the study is in places difficult to understand and not always stringent. I would like to point out one general difficulty that I had with the paper before I list a couple of specific comments.

*General comment:*

    2.1 You come to the conclusion that meteoric fragments can nucleate nitric acid hydrate crystals, however, there is unlikely to be sufficient input flux to explain observed crystal numbers in PSC. You further propose that acid processed meteoric smoke can also nucleate NAT, despite its small size. In contrast to meteoric fragments, observed crystal numbers in PSCs can be explained by the input flux of meteoric smoke. The first question I have is, why do you extend your previous setup from your study in 2018 by adding $H_2SO_4$ only for the experiments with meteoric fragments? Why don't you do this not also for meteoric smoke which seems to be much more relevant for PSC formation? And here comes again the question: Why is it not possible to measure MSP analogues in a ternary $H_2SO_4/HNO_3/H_2O$ solution? This would increase the value of this study considerably.

In fact, we did attempt to measure ternary solutions containing silica particles but found that the suspensions rapidly formed a gel. This was mentioned, but we have expanded our discussion of this in the methods section (line 195). Since it is thought that silicon is present in stratospheric droplets as discrete particles rather than throughout the droplet (Murphy et al., 2014), this gel formation cannot be atmospherically relevant, and is likely an artifact of the laboratory temperature. Further reduction of the temperature at which suspensions and are made is not practical without affecting solution concentrations e.g. by deposition of atmospheric water. We agree that this is a significant limitation of our experimental study. Other experiments such as cloud chamber studies may be better suited to studying the low temperature effect of $H_2SO_4$ on nucleation activity of meteoric smoke, although the low vapour pressure of $H_2SO_4$ at room temperature also makes these studies difficult. We have now added this suggestion for future work to at the end of section 4.6.

    2.2 The second request I do have: I would like to see equal figures for both types of meteoric material. I would like to compare meteoric fragments and meteoric smoke under the same conditions by looking at the same figures. Let's take Fig. 3 and Fig. 9: I would like to see the same figure with the same axis labels with the same axis numbers with the same caption and so on. The only difference should be the data.

*This is a reasonable request and we did endeavour to present the data in this way in our previous paper. However in this case, as a result of the different theoretical frameworks we use to analyse the two materials a like with like comparison is not possible until the full atmospheric conditions are used. We have added our limiting fragment concentration to Figure 11 to reflect this. Note that comparison of fraction frozen for different materials is rather misleading unless the same surface area of each heterogeneous nucleating material is included in solution, which is not the case here. Figure 9 is included to illustrate the process of fitting CNT parameters to our experimental data, which is only relevant to amorphous silica as an MSP analogue.*

*Specific comments:*

2.3 Introduction: Please revise your general PSC introduction completely. It contains the most important PSC facts but the details are not thoroughly written down. The selection of references could be better, too. Page 2, Line 36: I would add more citations, e. g. Manney et al. (2020), Wohltmann et al. (2020), Dameris et al. (2021). Lawrence et al. (2020) is particularly about the vortex strength. Page 2, Line 42: In my view, Wegner et al. (2012) is not appropriate here. Please search for a better reference related to denitrification. I would suggest Crutzen and Arnold (1986).
*Thank you for these suggestions: the text has been revised with these recommended citations added.*

2.4 Page 2, Line 43 ff: The sentence "In many clouds NAT is thought to nucleate on ice crystals, but it has been shown that crystalline nitric acid particles can sometimes form in conditions where ice is not thermodynamically stable (Mann et al., 2005; Tritscher et al., 2021)" contains a valuation which is probably not on purpose. It sounds like the majority of NAT particles is formed through the nucleation on ice crystals and only "some" NAT particles form via different pathways. I would express this in a more neutral way because this weighting might be wrong from today's perspective.
*The language of this section has been revised as suggested. Our intention is not to investigate the competition of these nucleation pathways, but rather to attempt to describe the nucleation process in the absence of ice.*

2.5 Carslaw et al. (1999) shows results from a model study stating that mountain-induced mesoscale temperature perturbations may be an important source of nitric acid hydrate particles in the Arctic. He does not claim that "all" clouds contain water ice. In 1999, he was still missing the vortex wide picture by satellite observations. Also the model resolution was quite coarse at that time (5 x 2 degree). Therefore I do not support your statement "though in some winters effectively all cloud could contain water ice" – in general and also not with this citation.
*This has been removed.*

2.6 Page 2, Line 50 ff: "Global models do not yet include a parameterisation of this activity based on laboratory measurements of reasonable heterogeneous nucleator surfaces, but rather have been tuned to observed NAT particle concentrations (Grooß et al., 2014; Voigt et al., 2005)." You cite Grooß et al. (2014), however, this paper contains a NAT parameterization based on the active site theory. Also Zhu et al. (2015, 2017a,b) describes the very detailed PSC scheme of WACCM. On the other hand, you cite Steiner et al. (2021) which is a global model. However, you cite it together with box models. Please have a look at Table 7 from Tritscher et al. (2021) to find out more about PSC parameterizations in models and revise this paragraph.
*This section has been thoroughly revised and expanded. The important message is that no model uses a parameterisation of heterogeneous nucleation in PSCs which is based on laboratory data, for good reason: none has been available until now.*

Whilst the formalism of the parameterisation in the work of Grooß (2014) and Hoyle (2013) are based on the laboratory work of Marcolli (2007), the key factors which describe the activity of the heterogeneous nucleus (the distribution of active sites); $\alpha_i$, $P_{pre}$ and $\gamma'$ were fitted to atmospheric observations. In addition, the particle size and number density which determines the total number of active sites were assumed.

2.7 Page 2, Line 55: Figure X1 → Figure 1

This has been corrected.

2.8 Figure 1: Your y-Axis (Altitude) goes down to 0, even below. The $H_2O$ concentrations in the troposphere are higher than 5 ppm! Please draw the lines for $H_2O$, $HNO_3$ and $H_2SO_4$ concentrations more carefully/exact.

This figure has been revised. Since these example concentrations are shown to indicate gas phase species which fragments can interact with above PSC altitudes, they have been removed below 20 km.

2.9 Within the introduction, I would like to see a clear definition of meteoric smoke and meteoric fragments. Please explain the differences between smoke and fragments such that a reader who is not as familiar with the topic and nomenclature as you are can easily understand it. This is important to get the main massage of your paper! It is also important to use this terminology through the whole paper. If you introduce an abbreviation, like MSP, please use this abbreviation everywhere and do not switch back to the write out version, this is confusing.

The introduction around line 60 has been expanded to include this. See also response to comment 1.6.

2.10 Page 8, Line 153: Could you please explain in more detail where the 0.5 wt% $H_2SO_4$ is coming from?! If I look at the corresponding plot (Fig. 12 in Carslaw et al., 1997), I assume a value higher than 0.5 wt%. Looking at Tab. 1 in Biermann et al. (1996), based on Carslaw et al. (1994), it would be at 190.0 K a value of 41.2 wt% for $HNO_3$ and 3.9 wt% for $H_2SO_4$. Even below the frost point, the concentration of $H_2SO_4$ is still at 2.5 wt%.

The reviewer is correct that this low concentration of $H_2SO_4$ is a rather unusual limiting case. It is only possible where the $HNO_3$ concentration is higher than the 10 ppbv used in the references mentioned here. An example set of conditions has been added to the text (line 184); these were used with the e-AIM model to calculate equilibrium droplet concentrations. Again, a chamber experiment where the concentration $H_2SO_4$ and $HNO_3$ in droplets evolves during cooling in a more realistic manner would be a logical next step, rather than trying to do many cold stage freezing experiments to explore the pertinent parameter space. Nevertheless, we have shown that meteoric fragments can still nucleate NAT in the presence of $H_2SO_4$.

2.11 Page 10, Figure 3, Caption: Please make one or two complete sentences at the beginning. "are data from (James et al., 2018)" change to James et al. (2018)

This caption has been revised.

2.12 Page 12, Figure 4: Please describe in the caption of the figure always the abbreviations used in the figure, here $n_s$ and $S_{NAT}$. It is more convenient for the reader to find it in the caption instead of searching the main text body.

This has been added to the figure caption.

2.13 Page 13, Line 230: Where does the $6 \times 10^{-6}$ cm$^{-3}$ comes from? Reference?

As well as expanding the discussion and adding reference around line 265, the caption of Figure 11 has also been revised for clarity.

2.14 Page 17, Line 318: 3:1 $H_2O$:$HNO_2$ change to $HNO_3$

This has been corrected.

2.15 Page 22, Line 413: Change 0.236 mJ m$^{-2}$ in 0.236 J m$^{-2}$

This has been corrected.
2.16     Page 25, Line 450f (Caption of Fig. 10): Change ppmv in ppbv $HNO_3$
This has been corrected.
2.17     My impression is that you need to check carefully when you use italic fonts. Sometimes, the unit is in italic fonts, sometimes the variable and sometimes none of both.
This has been checked and changes made throughout the document.
2.18     Please distinguish more carefully surface area and surface area density (e. g. y-axis Fig. 6 is surface area density).
This has been checked and changes made throughout the document.
2.19     Please use abbreviations contentiously throughout the manuscript and don't switch forth and back with using them or not.
This has been checked and changes made throughout the document.

*Additional references that were not listed in the paper itself:*

Biermann, U. M., Presper, T., Koop, T., Mößinger, J., Crutzen, P. J., & Peter, T. (1996). The unsuitability of meteoritic and other nuclei for polar stratospheric cloud freezing. Geophysical Research Letters, 23(13), 1693–1696. https://doi.org/10.1029/96GL01577

Carslaw, K. S., Luo, B. P., Clegg, S. L., Peter, T., Brimblecombe, P., & Crutzen, P. J. (1994). Stratospheric aerosol growth and HNO3 gas phase depletion from coupled HNO3 and water uptake by liquid particles. Geophysical Research Letters, 21(23), 2479–2482. https://doi.org/10.1029/94GL02799

Crutzen, P. J., & Arnold, F. (1986). Nitric acid cloud formation in the cold Antarctic stratosphere: A major cause for the springtime "ozone hole". Nature, 324, 651–655. https://doi.org/10.1038/324651a0

Dameris, M., Loyola, D. G., Nützel, M., Coldewey-Egbers, M., Lerot, C., Romahn, F., & van Roozendael, M. (2021). Record low ozone values over the arctic in boreal spring 2020. Atmospheric Chemistry and Physics, 21(2), 617–633. https://doi.org/10.5194/acp-21-617-2021

Manney, G. L., Livesey, N. J., Santee, M. L., Froidevaux, L., Lambert, A., Lawrence, Z., et al. (2020). Record-low Arctic stratospheric ozone in 2020: MLS observations of chemical processes and comparisons with previous extreme winters. Geophysical Research Letters, 47(16). https://doi.org/10.1029/2020GL089063

Marcolli, C., Gedamke, S., Peter, T., and Zobrist, B.: Efficiency of immersion mode ice nucleation on surrogates of mineral dust, Atmos. Chem. Phys., 7, 5081–5091, doi:10.5194/acp-7-5081-2007, 2007.

Murphy, D. M., Froyd, K. D., Schwarz, J. P., and Wilson, J. C.: Observations of the chemical composition of stratospheric aerosol particles, Quart. J. Royal Met. Soc., 140, 1269-1278, 2014.

Whale, T. F., Murray, B. J., O'Sullivan, D., Wilson, T. W., Umo, N. S., Baustian, K. J., Atkinson, J. D., Workneh, D. A., and Morris, G. J.: A technique for quantifying heterogeneous ice nucleation in microlitre supercooled water droplets, Atmos. Meas. Tech., 8, 2437–2447, https://doi.org/10.5194/amt-8-2437-2015, 2015.

Wohltmann, I., Gathen, von der, P., Lehmann, R., Maturilli, M., Deckelmann, H., Manney, G. L., et al. (2020). Near complete lo-cal reduction of Arctic stratospheric ozone by severe chemical loss in spring 2020. Geophysical Research Letters, 47. https://doi.org/10.1029/2020GL089547

Zhu, Y., Toon, O. B., Lambert, A., Kinnison, D. E., Bardeen, C., & Pitts, M. C. (2017). Development of a polar stratospheric cloud model within the community earth system model: Assessment of 2010 Antarctic winter. Journal of Geophysical Research: Atmospheres, 122(19), 418–510. https://doi.org/10.1002/2017JD027003

Zhu, Y., Toon, O. B., Lambert, A., Kinnison, D. E., Brakebusch, M., Bardeen, C. G., et al. (2015). Development of a polar tratospheric cloud model within the community earth system model using constraints on type I PSCs from the 2010–2011 Arctic winter. Journal of Advances in Modeling Earth Systems, 7, 551–585. https://doi.org/10.1002/2015MS000427

Zhu, Y., Toon, O. B., Pitts, M. C., Lambert, A., Bardeen, C., & Kinnison, D. E. (2017). Comparing simulated PSC optical properties with CALIPSO observations during the 2010 Antarctic winter. Journal of Geophysical Research: Atmospheres, 122, 1175–1202. https://doi.org/10.1002/2016JD025191

---

## Author Response (AR2)

Reviewer 1:

The authors have answered my questions/criticisms to my satisfaction as far as the present manuscript is concerned. The authors may want to amend/improve the manuscript as follows before submitting their final version:

Line 74: replace "polymerize" by "condense"

This change has been made.

Line 126: The relevant reaction for denitrification is ClO or BrO + $NO_2$ going to $ClONO_2$ or $BrONO_2$. It is thus $NO_x$ and not $NO_y$ that brings about the effect of lowering the rate of $Cl_2$ or BrCl re-formation in competition with $Cl_2$ or BrCl photolysis followed by rapid ozone destruction.

This has been corrected.

Line 193: "…not be important in the atmosphere". The authors do not advance an argument that would obviate the presence of amorphous silicon or silicates in aerosol particles. We simply do not know.

This is discussed further in a later Section 4.4 of the paper and reference has been added to that in the methods section. Several atmospheric observations support the presence of solid silica, without metallic counter ions, and are not consistent with the presence of gel in the atmosphere.

Line 330: Dotted line in Figure 6 indicates 0.20 rather than the indicated 0.25 sq micron $cm^{-3}$ in the Figure caption.

This has been corrected.

Figure 10: Red curve (broken line) is nowhere mentioned. What is it?

In this figure the value of the surface energy used in the model is represented by the line type, and the assumed atmospheric $HNO_3$ concentration by line colour. The figure caption has been edited to more clearly describe the data sets shown.

Reviewer 2:

Thanks for your revised manuscript. I can follow your argumentation as a response to my review. However, I have still two minor, technical points:

In the new manuscript, NAX has been introduced. Please check if this is necessary and if you use it properly. Just as an example: Voigt et al. (2005) is talking about NAT and not NAX (line 293). In general, don't switch back and forth if it is just terminology. It makes it more difficult to understand.

The editor's comment regarding α-NAT demonstrates that there remains interest in the field as to the differing phases of NAX which might form in the polar stratosphere, so we do think that it is important to distinguish when e.g. a previous study has measured nucleation of NAD as well as NAT. That said we had used NAX several times when referring to our atmospheric box model, where in this paper we have specifically made the assumption that NAT is nucleating, so those cases have been amended to specify NAT.

The usage of italic vs. upright fonts for variables is still confusing. I guess that this will be corrected by the journal. However, you need to correct at least your figures:

The overall rule is that symbols representing physical quantities (or variables) are italic, but symbols representing units, or labels, are roman (upright). You should not write whole words like "Temperature" in italic fonts.

We have removed italics from the names of variables where these were written out in figures.

Editor Comment:

Please comment on the polymorphism of NAT. Did you observe the nucleation of alpha-NAT or beta-NAT? This is an important question since the nucleation barriers for both phases and between them are rather different, see Weiss et al. 2016 and literature as quoted within.

Specific mention of α-NAT has been added to the text (line 385) along with the fact that as we only observe the phase which grows and melts, we have only very limited information about the phase which initially nucleates. Our experiments are designed to assess the amount of nitric acid hydrate nucleation in the atmosphere, rather than the phase chemistry. What we can say is that given literature data on the stability of possible phases, we see some nucleation events which could only be NAT, and most could not be water ice. We note that the central assertion of Weiss et al. (2016) is that α-NAT could nucleate on water ice given their similar crystal structure, however our experiments and analysis are targeted at atmospheric clouds which form in the absence of water ice. Since we see evidence of recrystallisation in some control experiments we are likely sensitive to it to some extent, and since we do not see it in heterogeneous experiments we do not believe that α-NAT is forming and converting to β-NAT, however we do not feel there is sufficient evidence to include this conjecture in the main text of the paper.

**Commented [A(J1):** Not sure we need this?